# Rad-cGAN v1.0: Radar-based precipitation nowcasting model with conditional Generative Adversarial Networks for multiple dam domains

Suyeon Choi[1], Yeonjoo Kim[1]

[1]Department of Civil and Environmental Engineering, Yonsei University, Seoul 03722, Korea

*Correspondence to:* yeonjoo.kim@yonsei.ac.kr

**Abstract.** Numerical weather prediction models and probabilistic extrapolation methods using radar images have been widely used for precipitation nowcasting. Recently, machine-learning-based precipitation nowcasting models have also been actively

developed for relatively short-term precipitation predictions. This study aimed to develop a radar-based precipitation nowcasting model using an advanced machine learning technique, conditional generative adversarial network (cGAN), which shows high performance in image generation tasks. The cGAN-based precipitation nowcasting model, named Rad-cGAN, developed in this study was trained with the radar reflectivity data of the Soyang-gang Dam Basin in South Korea with a spatial domain of 128×128 pixels, spatial resolution of 1 km, and temporal resolution of 10 min. The model performance was

evaluated using previously developed machine-learning-based precipitation nowcasting models, namely convolutional long short-term memory (ConvLSTM) and U-Net. In addition, Eulerian persistence model and pySTEPS, radar-based deterministic nowcasting system, used as baseline models.

We demonstrated that Rad-cGAN outperformed reference models at 10-min lead time prediction for the Soyang-gang Dam Basin based on verification metrics: Pearson correlation coefficient (R), root mean square error (RMSE), Nash–Sutcliffe

efficiency (NSE), critical success index (CSI), and fraction skill scores (FSS) at intensity threshold of 0.1-, 1.0-, and 5.0-mm $h^{-1}$. However, unlike low rainfall intensity, the CSI of high rainfall intensity in Rad-cGAN deteriorated rapidly beyond a lead time of 10 min; however, ConvLSTM and baseline models maintained the better performances. This observation was consistent with the FSS calculated at high rainfall intensity. These results were qualitatively evaluated using Typhoon Solik as an example, and ConvLSTM maintained relatively higher precipitation than the other models. However, for prediction of

precipitation area, Rad-cGAN showed the best results, and the advantage of cGAN method to reduce the blurring effect was confirmed through power spectral density (PSD). We also demonstrated the successful implementation of the transfer learning technique to efficiently train model with the data from other dam basins in South Korea, such as the Andong and Chungju Dam basins. We used pre-trained model, which was completely trained in the Soyang-gang Dam Basin. Furthermore, we

analysed the amount of data to effectively develop the model for the new domain through the transfer learning strategies

applying the pre-trained model using data for additional dam basins. This study confirmed that Rad-cGAN can be successfully

applied to precipitation nowcasting with longer lead times and using the transfer learning approach showed good performance

in dam basins other than the originally trained basin.

# 1 Introduction

Nowcasting is defined as a description of the current weather and then forecasting within few hours and is generally applied to mesoscale and local scales. Due to the increasing number of disasters in small spatiotemporal scales, nowcasting plays an important role in risk management (WMO, 2017). Therefore, the need for accurate precipitation nowcasting for early warning systems is increasing to reduce the damage caused by heavy rain, landslides, and flash floods.

Among the existing precipitation nowcasting models, numerical weather prediction (NWP), which performs rainfall prediction based on atmospheric physics equations, can generate high-resolution rainfall forecasts with long lead times. However, NWP has exhibited poor forecast performance with relatively short (0–2 h) lead times (Berenguer et al., 2012). Several studies have demonstrated that radar-based models based on the extrapolation method perform better than NWP, especially in the case of precipitation nowcasting with lead times of up to 6 h (Berenguer et al., 2012; Pierce et al., 2012; Renzullo et al., 2017; Imhoff et al., 2020). Additionally, the increased availability of high-resolution remote sensing observation data (e.g., radar) and computer resources has facilitated the development of advanced precipitation nowcasting models. For example, Ayzel et al. (2019) developed an optical flow-based precipitation nowcasting model called rainymotion, and Pulkkinen et al. (2019) developed a deterministic and probabilistic nowcasting application called pySTEPS, which have potential applications in several countries (Finland, Switzerland, the United States, and Australia). Both models were written in an open-source Python library. Furthermore, the blending technique, which combines NWP and radar-based models, has improved the precipitation nowcasting performance for short-term flood forecasting (Poletti et al., 2019; Hwang et al., 2020).

Recent availability of large amount of data and increased computational resources led to the development of radar-based models using machine learning techniques. Shi et al. (2015) developed a radar-based model with a convolutional long short-term memory (ConvLSTM) architecture that outperformed the optical flow-based model. They showed that ConvLSTM can capture the spatiotemporal correlation between input rainfall image frames, which are recorded every 6 min across Hong Kong. Several studies have shown that the ConvLSTM architecture can be successfully applied to the precipitation nowcasting model (Kim et al., 2017; Moishin et al., 2021; Sønderby et al., 2020; Jeong et al., 2021). Although the convolution neural network (CNN) does not have a structure to conserve temporal information, Agrawal et al. (2019) showed that a fully connected CNN called U-Net can make better predictions than traditional NWP models. Further studies (e.g., Ayzel et al., 2020; Trebing et al., 2021) also confirmed that the U-Net architecture can accurately predict precipitation.

In the field of computer science, the generative adversarial network (GAN) architecture (Goodfellow et al., 2014) showed remarkable performance in image-to-image tasks. Isola et al. (2017) demonstrated that the U-Net model with a conditional GAN (cGAN) approach called Pix2Pix can generate higher-quality images than the original U-Net model. Rüttgers et al. (2019) showed that typhoon tracks and cloud patterns over the Korean Peninsula could be successfully predicted using cGAN architecture with satellite cloud images. Also, Ravuri et al. (2021) developed a precipitation nowcasting model using a deep generative model inspired by the video GAN model (Clark et al., 2019). In the case study of convective cells over eastern

Scotland, using video GAN in the model improved the quality of precipitation forecasts significantly (Ravuri et al., 2021). These studies indicate that the performance of precipitation nowcasting models can be improved by advanced machine learning techniques. However, because machine learning is a data-driven technique, it will perform effectively for only trained data domains. Generally, it is vital to train from the beginning to develop a model for a new domain, and computation costs will be high even if new data are similar to old data. Thus, the models trained for one domain will be limited in their applications for multiple regions.

The aim of this study was to develop an advanced precipitation nowcasting model for multiple dam basins that can be applied as an early warning system. The decision-making process at upstream dams with regard to flood control, which is directly related to urban and rural water management, influences flood risk considerably. From such a dam management perspective, water level and inflow at dam sites are major factors to be considered, suggesting that increasing rainfall prediction accuracy over the entire dam basin is essential for effective flood management. To develop an advanced precipitation nowcasting model with good prediction performance for dam basins in general, we designed a model based on the cGAN approach (Rad-cGAN) for multiple dam domains of the Soyang-gang, Andong, and Chungju dam basins in South Korea. We trained the model using radar reflectivity data from the Soyang-gang Dam Basin for the summer season during 2014–2017 (provided by the Korea Meteorological Administration, KMA) and evaluated model performance using the 2018 data by comparing it with reference models of ConvLSTM, U-Net, and Eulerian persistence. We also used spectral prognosis (S-PROG) (Seed, 2003), which is a deterministic nowcast model in the pySTEPS library, for evaluation. We applied the transfer learning technique (Pan and Yang, 2009) that uses the previously trained model with cost-effective computation to train the model for the other two abovementioned domains. Five transfer learning strategies were compared to evaluate which was most effective for the Andong and Chungju Dam basins.

## 2 Materials and methods

### 2.1 Study area and Radar reflectivity data

We developed a precipitation nowcasting model for dam basins where an accurate rainfall forecasting system is essential for the estimation of urban water supply and flood prevention. The target domains were the Soyang-gang Dam Basin (D1), Chungju Dam Basin (D2), and Andong Dam Basin (D3) areas. These dams are multi-purpose and are located upstream of the major rivers of South Korea (Fig. 1).

The 1.5-km constant altitude plan position indicator (CAPPI) radar reflectivity data, provided by KMA, were used as input data for training and evaluation of our model. The map product represents the quality-controlled radar reflectivity composite (dBZ) of 11 weather radar stations across South Korea (Fig. 1a), with a size of 960×1200 pixels, spatial resolution of 1km, and temporal resolution of 10 min.

The radar composite data were cropped to 128×128 pixels, covering three target basins. Figure 1b shows the different topographical characteristics of each domain. Since topography (especially mountainous areas such as study domains) affects atmospheric conditions, such as temperature, humidity, air pressure distribution, and cloud formation, it directly or indirectly affects rainfall formation and distribution (Basist et al., 1994; Prudhomme and Reed, 1998). Consequently, data extracted from the three domains with different topographic characteristics would exhibit different rainfall patterns. We selected the available radar reflectivity data in summer (June–August, JJA) from 2014 to 2018 considering high intensity rainfall occurs in summer due to rainfall seasonality, a characteristic of our study domain. Data from 2014 to 2017 were used for training the model, and data from 2018 were used for evaluation (Table 1). For rapid and effective training, the raw radar reflectivity data (dBZ) were converted to grayscale (0–255), and the data range was scaled to 0–1 using the Min-Max scaler method (min-max values from training dataset). The predicted radar reflectivity data were converted into precipitation using the Z-R relationship (Eq. (1)) (Marshall and Palmer, 1948) to evaluate the rainfall prediction performance of the model.

$$Z = 200R^{1.6} \tag{1}$$

where Z is the radar reflectivity factor ($mm^6 m^{-3}$) and R is the rainfall rate ($mm\ h^{-1}$).

We used cropped radar reflectivity images (128×128 $km^2$) of the Soyang-gang Dam Basin to train and evaluate the proposed model (Rad-cGAN) and reference models (U-net, ConvLSTM, Eulerian persistence, and pySTEPS (S-PROG)). Furthermore, to reduce the edge effect caused by the fast Fourier transform (FFT), which is used for scale decomposition of the pySTEPS (S-PROG) nowcast (Pulkkinen et al., 2019; Foresti and Seed, 2014), we derived the pySTEPS results using 384×384 $km^2$ input data extended by 128 pixels on each side of the original input data (128×128 pixels).

## 2.2 Model architecture

### 2.2.1 Conditional generative adversarial network (cGAN) for image translation

GAN is a recently developed framework for training generators (e.g., CNN encoder-decoder) via an adversarial process. It consists of a generator ($G$) that produces the distribution of real data from random noise, and a discriminator ($D$) that classifies whether the input sample is from the generator or the original data distribution (Goodfellow et al., 2014). Furthermore, the cGAN framework uses additional conditions (e.g., input data of the generator) for training and can generate targeted outputs that suit specific conditions (Mirza and Osindero, 2014). For image translation tasks, when $G$ is trained to produce a targeted image ($y$) from input ($x$) with random noise ($z$), the objective of $D$ will try to maximize the loss function $\mathcal{L}_{cGAN}(G, D)$ while $G$ will try to minimize $\mathcal{L}_{cGAN}(G, D)$. This relation can be expressed as:

$$\min_{G} \max_{D} \mathcal{L}_{cGAN}(G, D) = \mathbb{E}_{x,y}[\log D(x, y)] + \mathbb{E}_{x,z}[\log(1 - D(x, G(x, z)))], \tag{2}$$

where losses were calculated as expected ($\mathbb{E}$) values. After simultaneously training $G$ and $D$, $G$ was trained to generate an output that cannot be distinguished from real data ($y$) by $D$, which was trained in an adversarial manner to detect the fake

image from $G$. Isola et al. (2017) showed that combining the traditional pixel-wise loss with cGAN loss can improve the quality of output images. To generate sharp and realistic images, the $L1$ loss function $\mathcal{L}_{L1}(G)$ was used as the traditional loss (Eq. (3)).

$$\mathcal{L}_{L1}(G) = \mathbb{E}_{x,y}[\|y - G(x,z)\|_1] \tag{3}$$

By adding the traditional loss with a weight $\lambda$ to the cGAN loss, the final objective was obtained (Eq. (4)).

$$G^* = arg \min_G \max_D \mathcal{L}_{cGAN}(G,D) + \lambda\mathcal{L}_{L1}(G) \tag{4}$$

In this study, we developed a radar-based precipitation nowcasting model using a cGAN framework. Recently, research on weather prediction using cGAN, an advanced machine learning approach, has been conducted extensively (e.g., Rüttgers et al., 2019; Ravuri et al., 2021). For example, Ravuri et al. (2021) proposed a generator consisting of two modules; conditioning stack (using CNN to extract representation of input); and sampler (using ConvGRU to generate prediction). The model, which used ConvGRU, could observe spatiotemporal changes of inputs such as ConvLSTM, and attempted to improve performance by extracting features from different spatial dimensions and deriving the results. Whereas the generator used to predict future radar map, the discriminator used a dual architecture that distinguishes the real and generated frames, to ensure both temporal and spatial consistency. Unlike the model proposed by Ravuri et al. (2021), our model adopts a U-net architecture that uses a CNN layer in image generation based on the underlying Pix2Pix model; the architecture exhibits outstanding performance in image-to-image translation tasks (Isola et al., 2017). Also, we considered only spatial consistency in the PatchGAN discriminator, which distinguishes images for each N×N patch (N can be smaller than the full size of the image). U-net-based precipitation nowcasting model has previously demonstrated performance superior to that of a traditional radar-based precipitation nowcasting model that uses optical flow (Ayzel et al., 2020). Therefore, here, we apply the basic cGAN methodology to the U-net structure to improve performance and confirm the applicability of the transfer learning methodology to multiple dam domains.

### 2.2.2 Generator

Figure 2a shows the generator using U-Net architecture (a detailed description of U-Net is provided in Sect. 2.3.3). The model consists of nine convolutional layers, two max-pooling layers, two up-sampling layers, and an output convolutional layer. Each convolutional layer, except for the output layer, is composed of the following operations: 3×3 2D convolution with zero-padding, batch normalization, and activation function of ReLU. In the contracting part of the generator, a 2×2 2D max-pooling operation was used to down-sample the input images. To prevent overfitting, a dropout layer with a rate of 0.5 was applied after the pooling and convolutional layers of the expanding part of the model (Srivastava et al., 2014). A 2×2 2D up-sampling operation was further applied in the expanding part after skip connection to increase the resolution of featured images that contain both high- and low-level information. Finally, the output convolutional layer had a 1×1 2D convolution that used a linear function for activation to obtain future prediction of the radar reflectivity image.

**2.2.3 Discriminator**

PatchGAN from the Pix2Pix model was used as the discriminator (Fig. 2b). As in cGAN, the input pair of the discriminator consists of historical radar reflectivity data (i.e., input of the generator) and future radar reflectivity data. The discriminator classifies real image pairs (input of generator and ground truth image) as 1 and fake image pairs (input and generated image from generator) as 0 (Mirza and Osindero, 2014). Particularly, PatchGAN only penalizes the structures over a certain scale of image patches; therefore, the discriminator classifies whether the N×N patch in the input pair is real or fake. This patch represents the receptive field, which is the region in the input image that is used to measure the associated feature of the output layer. Consequently, the size of the patch (N) was determined based on the structure of the entire discriminator (e.g., number of layers, nodes, filter size, paddings, and strides), and it increased as the model deepened. We constructed a discriminator model with a 34×34 patch size through hyperparameter tuning. The model consists of three convolutional layers and an output layer. The first two convolutional layers were composed of 4×4 2D convolution with strides of two and zero-padding, batch normalization, and ReLU activation function, which was leaky and had a 0.2 slope. The third convolutional layer had the same configuration as the previous layers, except that its stride was 1. To distinguish the input pair in the image form, the output layer consisted of 4×4 2D convolution with zero-padding and sigmoid activation functions. To train the discriminator as a classifier, we manually generated the training datasets consisting of the input image pairs and the target images, with spatial dimensions of 32×32 filled with 1 (for real image pairs) or 0 (for generated image pairs). Therefore, each pixel of the output estimates the probability that the discriminator determines each patch of the input pair as the real one.

**2.2.4 Optimization procedure**

Before proceeding with training to optimize the model for the input data, hyperparameter tuning is required to determine the most optimal model structure and training settings. We selected the following hyperparameters: number of layers, number of hidden nodes, convolution filter size, patch size, batch size, and learning rate. To select the appropriate hyperparameter combination, the model for each combination was trained using radar data from 2014 to 2016 (June to August) and data from 2017 (June to July). Subsequently, using data from 2017 (August), the mean absolute error (MAE) and critical success index (CSI) (at an intensity threshold of 0.1 mm/h) were calculated to obtain the optimal combination of hyperparameters. Based on the tuning results, the MAE range was 0.45–47.66 and the CSI range was 0.0–0.83, and the results confirmed that hyperparameters influence model performance considerably. Based on the combinations that performed optimally, we determined the model structure and training settings.

To optimize Rad-cGAN, the training procedure suggested by Isola et al. (2017) was adopted. First, we compared the results using a total of four and six consecutive radar reflectivity images to determine the input historical data length. As a result of 10 min precipitation prediction at the Soyang-gang Dam site, in the case of CSI (at the rainfall intensity of 0.1 mm h$^{-1}$), the case of using six historical data was slightly better than case of using four data, but in R, RMSE, and NSE, the results of using four data were better. Through this, samples that consisted of four consecutive radar reflectivity images (*t-30*, *t-20*, *t-10* min, and *t*) and the image at *t+10* min were selected. Subsequently, a training sample for the discriminator was created by adding labels to classify whether the samples were real (image at *t+10* min from observation) or fake (*t+10* image from the generator) pairs. Then, the parameters of the discriminator were updated using the minibatch stochastic gradient descent (SGD) method for one step. Binary cross-entropy was used as a loss function, and the ADAM optimizer (Kingma and Ba, 2015) with a learning rate of 0.0002 and momentum parameters $\beta_1 = 0.5$ and $\beta_2 = 0.999$ was applied. Afterwards, the generator was trained for one step to optimize Eq. (4). Binary cross-entropy was used as $\mathcal{L}_{cGAN}$ for the discriminator to classify the generated image into a real image. Additionally, $\lambda$ of the traditional pixel-wise $L1$ loss was set to 100. The minibatch SGD and ADAM optimizer were applied to train the generator with the same setting as the discriminator. Both the procedures for updating the parameters of the discriminator and generator were run simultaneously during one epoch. Our model was trained using 600 epochs, with a batch size of 8. To achieve the optimal model, we applied an early stopping technique that stops the training model when the loss stops improving. The loss metric was defined as the generator loss based on 100 validation samples randomly sampled from the training dataset that was not used to train the model. To monitor the loss, we set patience to 30 epochs and saved the model when the loss improved. The model architecture was written in Python (https://www.python.org/) using the Keras deep learning application (https://keras.io/). The entire procedure for training and all the experiments for evaluation were run on a computer with a single NVIDIA Tesla V100 GPU.

## 2.3 Reference models

The performance of the Rad-cGAN model was compared with and validated using reference models that include two baseline and two machine-learning models. We used the Eulerian persistence model (hereafter referred to as persistence), a traditional radar-based rainfall prediction model, as the baseline model. This model assumes that rainfall prediction at any lead time is the same as the rainfall in the forecast time. This is a simple but powerful model for predicting short-term precipitation. Additionally, a deterministic S-PROG nowcast from the pySTEPS library was used as another baseline model (Sect. 2.3.1). We used ConvLSTM (Sect. 2.3.2) and U-Net (Sect. 2.3.3), which are the common basic structures for machine learning-based nowcasting models, as reference models for comparison.

**2.3.1 PySTEPS**

PySTEPS (Pulkkinen et al., 2019) is an open-source and community-driven Python framework for radar-based deterministic

and probabilistic precipitation nowcasting, and is considered a strong baseline model (Imhoff et al., 2019; Ravuri et al., 2021).

In this study, deterministic S-PROG (Seed, 2003) nowcast from the pySTEPS library was used as the benchmark model.

To predict precipitation, we input the precipitation images (unit: dBR) transformed from four consecutive radar reflectivity

images (from *t-30* to *t*), which were the same as the input of the Rad-cGAN model, based on the Z-R relationship (Eq. (1)).

Additionally, the transformed precipitation was used to estimate the motion field, which was used together with precipitation

as input data in the model. Future precipitation at a lead time of up to 90 min for the test period (JJA, 2018) was generated

from the results of the S-PROG nowcasts. The source code of pySTEPS is available at GitHub repository

(https://pysteps.github.io, last accessed: 23 May 2022).

**2.3.2 ConvLSTM**

LSTM is a special case of recurrent neural networks (RNNs) and is widely used in temporal sequence predictions (Hochreiter

and Schmidhuber, 1997). Sutskever et al. (2014) proposed an LSTM encoder-decoder framework for sequence-to-sequence

problems, which consists of concatenated LSTMs for the input and output sequences. Based on this model, Shi et al. (2015)

developed a ConvLSTM network that can be applied to spatiotemporal sequence prediction, such as radar-based rainfall

prediction. To handle spatiotemporal sequences, a convolution operator was used in state-to-state and input-to-state transitions.

The ConvLSTM model was shown to outperform the traditional optical flow-based precipitation nowcasting model. Recent

studies have shown that the ConvLSTM model can be successfully applied to predict future radar-based precipitation (Kim et

al., 2017; Moishin et al., 2021).

We designed a ConvLSTM model that uses four radar reflectivity image frames (*t-30*, *t-20*, *t-10* min, and *t*) as input to predict

future frames at time *t+10* min, which is similar to input and output of Rad-cGAN. The model consists of three ConvLSTM

layers and an output layer. Each ConvLSTM layer contains 64 hidden states and 3×3 kernels. A 3D convolutional layer with

a linear activation function was used as the output layer. The hyperparameters of the ConvLSTM model (i.e., number of layers,

number of nodes, convolution filter size, batch size, and learning rate) were tuned using a procedure similar to that applied in

Rad-cGAN (Sect. 2.2.4). To optimize the model, we used the mean squared error as the loss function and applied the ADAM

optimizer (learning rate 0.002 and momentum parameters $\beta_1 = 0.9$ and $\beta_2 = 0.999$). We trained the model using 600 epochs

(early stopping applied) with a batch size of 32.

**2.3.3 U-net**

U-Net-based precipitation nowcasting models efficiently predict future precipitation using historical data, even though U-Net does not have a structure, such as RNN, that preserves temporal information (e.g., Ayzel et al., 2020; Trebing et al., 2021). U-Net was developed by modifying the fully convolutional network (Long et al., 2015), and performed well in image segmentation tasks (Ronneberger et al., 2015). This model architecture consists of two parts: a contracting network that captures the context of the input images and an expanding network that increases the resolution of features from the contracting network.

The contracting network follows the usual CNN, which consists of convolution and max-pooling layers. Each convolution layer is composed of convolution, batch normalization, and activation operations. Batch normalization is used to prevent gradient vanishing or exploding problems and can effectively increase the convergence speed (Ioffe and Szegedy, 2015). The max-pooling operation is applied for down-sampling after convolution of the input image. Through this process, the output of the contracting network can incorporate the features of the input image. The expanding network consists of the up-sampling and convolution layers. Before applying the up-sampling operation, the skip connection is applied between each layer of the contracting network and the layer of the expanding network to prevent gradient vanishing and share the low-level information of the input data (Simonyan et al., 2015). The convolution layers of the expanding and contracting networks follow the same operation.

As the reference model, hyperparameters for the U-net structure (number of layers, number of nodes, and convolution filter size) were set to be equivalent to those of Rad-cGAN (Sect. 2.2.2), and hyperparameters related to training settings (batch size and learning rate) were tuned using procedures similar to those of Rad-cGAN (Sect. 2.2.4). To optimize the model, $L1$ loss and ADAM optimizers were used as in the case of ConvLSTM (Sect. 2.3.2). The model was trained using 600 epochs with early stopping and the batch size set to 8.

**2.4 Experiments for evaluating model's prediction skills**

**2.4.1 Performance evaluation**

The model was trained using data from the summers (June–August) of 2014–2017 and its precipitation nowcasting capacity was assessed using data from the summer of 2018. To predict radar reflectivity data 10 min ahead, four latest radar reflectivity data (*t-30*, *t-20*, *t-10* min, and *t* min; *t* being the forecast time) were used as input data. The model can generate multiple samples (No. of samples, 128, 128, 1) corresponding to the number of samples of the past four consecutive input data (No. of samples, 128, 128, 4). To predict beyond the 10 min lead time, we used the prediction data at *t+10* min as the latest input data. Using this recursive process, predictions were obtained at a lead time of >10 min. Because the model predicts the radar

reflectivity after 10 min using past consecutive radar images, we first evaluated the model performance at a lead time of 10 min. This allowed us to confirm the prediction tendency of our model and other reference models while performing precipitation nowcasting. Furthermore, to assess the applicability of our model to an actual early warning system that needs to

ensure at least one hour of lead time, the predictive skill was evaluated for >10 min of lead time using the recursive process. We measured the verification metrics (see below) using rainfall prediction, converted from the radar reflectivity (Eq. (1)), at a lead time of up to 90 min to confirm the forecasting time in which the model ensured sufficient performance.

We evaluated the model performance of the entire dam basin since the effect on water level by rainfall over the dam basin is a major factor in the decision-making process for dam management (Fig. 1b). To evaluate the entire domain, the verification

metrics were calculated with increasing lead time for all pixels of the predicted image. Additionally, to qualitatively evaluate the entire domain, we compared the resulting precipitation images obtained using data at a certain forecast time. We set the forecast time at 23 August 2018, 17:50 UTC, when Typhoon Soulik, which landed on the Korean Peninsula from 23 August 2018, 12:00 UTC to 24 August 2018, 03:00 UTC, started affecting the Soyang-gang Dam Basin.

Several metrics were used for model evaluation: Pearson correlation coefficient (R), root mean square error (RMSE), Nash–

Sutcliffe efficiency (NSE), CSI, fractions skill scores (FSS). As the collinearity between actual rainfall and predicted rainfall increases, the explanatory power of the rainfall simulated by the model increases, so that the performance of the model can be illustrated by strong positive linear relationship between predictions and observations. Hence, we confirmed that the model exhibits better performance when R (Eq. (5)), calculated based on the model prediction and observation, is closer to 1. To verify the precision of the model, the RMSE (Eq. (6)) between prediction and observation was used. Since we propose a

precipitation nowcasting model, NSE, which is widely used to assess hydrologic models, can be used as a goodness-of-fit index for it (McCuen et al., 2006) (Eq. (7)).

$$R = \frac{\sum_{i=1}^{N}(O_i - \bar{O})(P_i - \bar{P})}{\sqrt{\sum_{i=1}^{N}(O_i - \bar{O})^2 \sum_{i=1}^{N}(P_i - \bar{P})^2}} \tag{5}$$

$$RMSE = \sqrt{\frac{\sum_{i=1}^{N}(O_i - P_i)^2}{N}} \tag{6}$$

$$NSE = 1 - \frac{\sum_{i=1}^{N}(O_i - P_i)^2}{\sum_{i=1}^{N}(O_i - \bar{O})^2} \tag{7}$$

where $\bar{O}$ and $\bar{P}$ are the means of observation and prediction, respectively, $O_i$ and $P_i$ are the observed and predicted precipitation, respectively, in the $i$th time of the data period, and $N$ is the total number of data for the entire period.

We used the CSI (Eq. (8)), which is a measure of categorical forecast performance, to verify the model accuracy for precipitation event detection.

$$CSI = \frac{hits}{hits + false\ alarms + misses} \tag{8}$$

where *hits* (correct event forecasts), *false alarms* (incorrect event forecasts), and *misses* (missed events) are defined by a contingency table (Table 2). Also, FSS can spatially verify model performance by comparing fraction of grid points of prediction and ground truth, which exceed certain rainfall intensity thresholds within the neighborhood (Eq. (9)).

$$FSS = 1 - \frac{\sum_{i=1}^{n}(P_p - P_o)^2}{\sum_{i=1}^{n}P_p{}^2 + \sum_{i=1}^{n}(P_o)^2} \tag{9}$$

where $P_p$ and $P_o$ are the fractions of prediction and observation, respectively, calculated by specific thresholds in neighborhood size. For calculating CSI and FSS, we selected several intensity thresholds, including 0.1-, 1.0-, and 5.0 mm h$^{-1}$, and for FSS, we used neighborhood sizes of 1-, 5-, and 15 km. Additionally, we calculated the radially averaged power spectral density (PSD) of predictions and observations to assess blurring effect of predicted image by models.

To calculate each verification metric, all metrics for each pixel in the dam basins were calculated and averaged over the data period (No. of samples).

### 2.4.2 Experiments for transfer learning among different domains

Since the machine learning model relies on input data as a data-driven model, training on the corresponding new data must be conducted from the beginning to develop a model for a new domain, which is also applicable for our precipitation nowcasting model for a new dam basin with different meteorological, environmental, and geographical characteristics (Fig. 1b). However, because this method is time-consuming and computationally expensive, we applied a transfer learning approach that can be efficiently used to train models with multiple dam basins.

Transfer learning is a machine learning technique that uses knowledge and skills from pre-trained models to train a model for new datasets (Pan and Yang, 2009). This method is often used when the size of the provided dataset is insufficient for training and is also used to train the models for the new dataset due to its lower computational cost than that of training from scratch. The general training strategies of transfer learning are determined by the data size and similarity between the new and original data. For example, if the new dataset is similar to the dataset of the pre-trained model, the new model only fine-tunes for higher layers that learn specific features of the input data and freeze the lower layers that capture the general features. Fine-tuning uses a smaller learning rate (e.g., ~1/10$^{th}$ of the original learning rate) and is one of the most effective ways to transfer knowledge. Several studies have shown that the transfer learning approach performs successfully well in image classification tasks (Krizhevsky et al., 2012; Simonyan and Zisserman, 2015; He et al., 2016). In the GAN approach, the discriminator acts similar to the classifier of the image classification task. Wang et al. (2018) reported that fine-tuning both the generator and discriminator resulted in good performance, but the overfitting was a frequent issue that must be considered. Subsequently, Mo et al. (2020) proposed a strategy that works only on the discriminator called FreezeD, which freezes the lower layer of the discriminator and only fine-tunes the upper layers.

We used transfer learning to train our model for different dam basins, i.e., Andong and Chungju, with a pre-trained model that was completely trained by data from the Soyang-gang Dam Basin. In addition, in existing papers that have successfully applied the transfer learning strategies, it was used to develop a model for a new domain using a pre-trained model based on vast data. Consequently, we used the pre-trained model with Daecheong Dam, Juam Dam, and Yongdam Dam basin data, in addition to Soyang-gang Dam data, to assess the amount of data required to develop a model for a new dam domain. The selected strategies

were inspired by a previous approach of transferring GAN (Wang et al., 2018; Mo et al., 2020). We formulated two strategies for each pre-trained model. First, the weights of the pre-trained generator were frozen and used directly in new dam domain (Cases 2 and 4). Next, the weights of the pre-trained generator were fine-tuned (1/10$^{th}$ of the original learning rate) and the discriminator trained from the scratch (Cases 3 and 5). In addition, the entire model was trained for the new domain (Case 1) (Table 3a). The model was trained for the Chungju and Andong Dam domains, separately, using the five strategies (Table 3b).

To determine the best strategy for training different dam domains, we estimated the performance at the 10-min lead time at each dam domain (Fig. 1b). Additionally, we compared the predictive skill of each strategy at the lead time of up to 90 min by using the recursive process.

## 3 Results and discussion

### 3.1 Domain-averaged model performance for the Soyang-gang Dam Basin

To apply our model in an early warning system, the prediction performance upstream of the dam should be sufficient. Hence, the verification metrics were calculated for each grid cell in the entire domain. First, we evaluated the performance of our model for the predicted precipitation at a lead time of 10 min at the Soyang-gang Dam Basin during the summer of 2018 (Table 4). As the general criterion for evaluating hydrological models, when R and NSE are $\geqslant$ 0.5, the model has acceptable

performance (Moriasi et al., 2007). In addition, Germann and Zawadzki (2002) suggested that the threshold of predictability is $1/e \fallingdotseq 0.37$, assuming that the verification metrics follow the exponential law. According to the standard, the mean values of each metrics of Table 4 shows that the machine learning-based models performed well as precipitation nowcasting models (R >0.5, NSE >0.5, CSI >0.5). Among them, Rad-cGAN outperforms the other reference models for almost all the verification metrics (Table 4). Particularly, Rad-cGAN shows improvements in the CSI values at different rainfall intensities (0.1-, 1.0-,

and 5.0-mm h$^{-1}$) by 0.55 %, 10.10 %, and 123. 50 %, respectively, compared with the model results using U-Net, confirming that the cGAN approach can mitigate the tendency to underestimate precipitation. However, by comparing with ConvLSTM and baseline models (especially for pySTEPS), Rad-cGAN performs poorly for highest rainfall intensity (5.0 mm h$^{-1}$).

We predicted the precipitation at the lead times of up to 90 min by using the recursive process, and their performance in all the grid cells were presented through boxplots for each lead time in all the models (Fig. 3). By comparing the median values,

Fig. 3 shows average increases of 9.02 % and 17.87% for the R of Rad-cGAN at overall lead times compared to those of U-Net and ConvLSTM, respectively, which indicate improved precipitation prediction capacity for the entire domain However, in the cases of RMSE and NSE, Rad-cGAN performs slightly worse than ConvLSTM, with an average increases over median values of 1.90% in RMSE and a decrease of 7.67% in NSE over the entire lead time (Fig. 3).

Moreover, according to the CSI value at the intensity of 0.1 mm $h^{-1}$, our model preserves its predictability performance (>1/e)

for the entire lead time, indicating that it can be applied to predict precipitation at lead times of >90 min. By comparing with previous studies, the lead time for the CSI at the intensity of 0.1 mm $h^{-1}$ >0.5 was up to 90 min with ConvLSTM in this study while CSI at the intensity of 0.5 (not 0.1) mm $h^{-1}$ >0.5 was up to 40 min with ConvLSTM-based nowcasting model for Hong Kong region (Shi et al., 2015). Also, Ayzel et al. (2020) showed that the U-net-based model preserved performance (CSI at an intensity of 0.125 mm $h^{-1}$ >0.5) at a lead time of >60 min in Germany, whereas the performance of our model with similar CSI

(0.1 mm $h^{-1}$ > 0.5) remained up to 90 min. Hence, we confirm that the reference model was sufficiently trained to be used for comparison with our model. The result indicates that Rad-cGAN has reliable performance in precipitation nowcasting for relatively light rain (rainfall intensity of 0.1 mm $h^{-1}$). However, in the case of CSI at intensities of 1.0- and 5.0 mm $h^{-1}$, although Rad-cGAN maintains a good performance compared to that of U-net, the performance rapidly deteriorates as the lead time increased. Unlike Rad-cGAN and U-net, ConvLSTM and baseline models record low CSI under low-intensity rainfall;

however, Fig. 3 shows that relatively high levels of performance are maintained under higher rainfall intensity. The results can also be confirmed through the FSS of each model (Fig. 4). When comparing Rad-cGAN and U-net, as lead time and rainfall intensity increase, both models decrease FSS; however, Rad-cGAN model exhibits superior performance. However, ConvLSTM and pySTEPS have the relatively high FSS values under high rainfall intensity compared to those of the other two models.

Thus, we observe a tendency to underestimate the prediction of high-intensity precipitation in all models, including Rad-cGAN (Figs. 3 and 4). A similar observation was made using ConvLSTM (Kumar et al., 2020), wherein significant errors occurred in precipitation prediction (>20 mm $h^{-1}$). This may be attributed to data imbalance, which is a common issue in machine-learning studies (Wang et al., 2016). Data imbalance occurred in this study because, unlike low-intensity precipitation (<5 mm $h^{-1}$), high-intensity precipitation rarely occurs during the training and testing periods.


**3.2 Spatial model performance for the Soyang-gang Dam Basin**

To better understand model performance with increasing lead time, we predicted precipitation for lead times of 10, 30, 60, and 90 min for a specific forecast time on 23 August 2018, 17:50 UTC, when Typhoon Soulik began affecting the Soyang-gang Dam Basin (Fig. 5). We observe that with an increase in lead time, the model performance deteriorates due to the blurring

effect of the predicted image, which is an issue reported in previous machine-learning-based nowcasting models (Ayzel et al.,

2020; Shi et al., 2015). Despite the smoothing trend, Rad-cGAN produces qualitatively better results than those of the other reference models (Fig. 5). The bias of prediction (observation-prediction) of Rad-cGAN at 90-min lead time ranges from -1.97 to 19.68 (mean = 0.83 mm h$^{-1}$), indicating that our model alleviates the underestimation of precipitation compared to U-Net, whose bias ranges from -0.30 to 20.33 (mean = 1.04 mm h$^{-1}$). The results support the improvement in Rad-cGAN verification

metrics compared to those of U-Net (Figs. 3 and 4). Furthermore, in the case of ConvLSTM, the mean bias of 0.86 mm h$^{-1}$ under a 90-min lead time prediction showed that ConvLSTM was less prone to underestimation when compared with U-Net. However, the 90-min rainfall prediction by ConvLSTM is recorded to be approximately 0 mm h$^{-1}$ in areas with an observation of ~5–10 mm h$^{-1}$, indicating that it predicts precipitation to be close to zero in most areas with increasing lead times. As Fig. 5 illustrates, ConvLSTM does not define the boundary of the entire area well but maintains high intensity rainfall compared

to the other two models, which causes the CSI difference depending on the rainfall intensity of ConvLSTM (Figs. 3 and 4). Figure 6 shows the PSD for each result in Fig. 5. Based on Fig. 5, all models exhibits a blurring effect compared to the ground truth. However, when comparing U-net and Rad-cGAN, Rad-cGAN has slightly lower blurring effect (Fig. 6). This is because a sharper image can be generated when cGAN is applied to the U-net structure (Isola et al. (2017)), which shows that the cGAN technique was successfully applied by our model. Therefore, based on the overall verification metrics, we conclude that

Rad-cGAN has the optimal prediction performance in nowcasting and prediction of spatial patterns of movement of precipitation.

PySTEPS shows poor performance (Fig. 5) compared to previous studies (e.g., Imhoff et al., 2020) in the verification metrics (Table. 4 and Figs. 3-4). The overall prediction performance degrads particularly because the precipitation area near the edge of the basin is not predicted. To better understand this side effect, we reran pySTEPS and Rad-cGAN with the extended data

of 384×384 pixels. Compared to the predictions in Fig. 5, the typhoon event (Fig. 7) shows that using the extended area reduces the edge effect of pySTEPS and properly maintains high rainfall intensity, thereby improving the performance. Moreover, the average R and CSI (at the highest rainfall intensity of 5.0 mm h$^{-1}$) for the 10-min precipitation prediction during the entire test period is calculated as 0.77 and 0.38, respectively, indicating that the performance improves quantitatively compared to the previous results (R=0.70 and CSI=0.32). Additionally, the prediction performance of typhoon event improves using the

extended area in Rad-cGAN (Fig. 7), and the average R and CSI (at the rainfall intensity of 5.0 mm h$^{-1}$) in the 10-minute rainfall forecast for the entire test period improves from 0.79 to 0.80 and from 0.18 to 0.37, respectively. Both models show improved performance using extended area, but considering the applicability of the model to real-world problems with limited data availability, we conclude that, unlike pySTEPS, Rad-cGAN is more efficient in rainfall prediction without considering the edge effects of the spatial size of the input domains.


### 3.3 Performance with transfer learning at different dam domains

To develop a precipitation nowcasting model for multiple dam basins (Andong and Chungju Dam Basins) other than Soyang-gang Dam Basin, we proposed to not only retrain our model with data from new dam basins (Case 1) but also apply efficient transfer learning methodology (Cases 2-5; Sect. 2.4.2).

First, we evaluated whether the transfer learning method could be effectively applied to the new domain (Cases 2 and 3) using a pre-trained model only for one domain (i.e., Soyang-gang Dam Basin). Table 5a shows the performance of each case with model predicted precipitation at 10-min lead time at the Andong Dam Basin. The results show that most of the verification metrics in Case 3 and Case 2 perform better than those in Case 1. In Case 2, which uses all the parameters of the generator from the pre-trained model, an NSE of 0.56 is achieved, which is closest to the NSE of the pre-trained model (0.54) with data

from the Soyang-gang Dam Basin. This is consistent with the verification metrics results at lead times of up to 90 min (Fig. 8).  Based on the median of R, Case 2 maintains the predictive performance (> 1/e) up to about 80-min lead time (Fig. 8a). Especially for CSI at higher rainfall intensities, Case 2 shows better performance than Case 1 overall lead times (Fig. 8d-f). Hyperparameter tuning would have had a significant impact on the results where Case 2 performs better than Case 1. Unlike the pre-trained model, which confirmed that model optimization and generalization were completed through the

hyperparameter tuning process, in case 1, we did not proceed with hyperparameter tuning for the new domain. Although the new domain has properties similar to those of the pre-trained domain, minor changes in hyperparameters also result in differences in performance, so that optimization and generalization of the model (Case 1) were less comprehensive than in the pre-trained model, resulting in relatively poor performance. However, in Case 3, the performance is lower than that of the other strategies. This is because of performance degradation due to overfitting during fine-tuning the pre-trained parameters.

High similarity between the two datasets of the Andong and Soyang-gang dams may be the reason for a major performance degradation of the transfer learning using the fine-tuning method (Wang et al., 2018).

For the Chungju Dam Basin, we trained the model using the same methodology used for the Andong Dam Basin. Among the Cases 1, 2 and 3, transfer learning cases (Cases 2 and Case 3) perform better than Case 1 for the overall verification metrics, especially for CSI at higher rainfall intensities (Table 5b). Additionally, when the lead time is increased by up to 90 min, Cases

2 and 3 show better performance than Case 1 for the entire lead time (Fig. 9). From the median values of CSI at the 0.1 mm h$^{-1}$, all three cases preserve sufficient performance (CSI >1/e) at a lead time of up to 90 min. However, as a result of comparing CSI with higher rainfall intensity (1.0 mm h$^{-1}$), it is confirmed that the performance in Case 2 only maintains predictive performance up to 30 min (Fig. 9e). Through these results, the Case 2 is successfully applied to Andong Dam and Chungju Dam among the transfer learning methodologies using a pre-trained model for the Soyang-gang Dam Basin.

Considering the advantages of transfer learning that can be effectively applied when data for new domains is insufficient over pre-trained domains, we evaluated the results of using pre-training models that had been trained for additional dam basins: Daecheong Dam, Juam Dam, and Yongdam Dam basins in addition to Soyang-gang Dam (Cases 4 and 5). Through the R for

10 min precipitation prediction at dam domains, Case 2 and 3, which used pre-trained model with Soyang-gang Dam Basin, show better performance for both Andong Dam and Chungju Dam (Table 5), but as the lead time increased to 90 min, Cases 4 and 5 maintain better performance up to 90 min (Figs. 8a and 9a). This trend is notable in RMSE and NSE. In addition, at CSI values of higher rainfall intensity, Cases 4 and 5 outperform other strategies at longer lead time for both Andong Dam and Chungju Dam (Figs. 8e-f and 9e-f). Since various and numerous data can solve the problem of data imbalance that causes underestimation of the model, the CSI value is good even at high rainfall intensity (Wang et al., 2016). In addition, Fang et al., (2022) showed that models trained through diverse and numerous data on multiple regions can also learn about the characteristics that contribute to regional differences and are more effective in predicting extreme events and future trends. These results show that the diversity and amount of data have no significant effect on the short-term prediction of low rainfall intensity but are very important in resolving model underestimation and improving prediction accuracy for heavy rainfall.

## 4 Conclusions

In this study, our aim was to develop a model that could be applied at each flood control center by focusing on developing a model for a dam basin. We developed a rainfall prediction model that could perform sufficiently well with a relatively simple structure and low computational costs, and evaluated the applicability of the transfer learning technique to facilitate its application in multiple dam basins. The proposed model could be used for rainfall-runoff modeling in dam basins in future work. To develop radar-based precipitation nowcasting model, we applied a cGAN approach based on the U-net architecture. The model architecture was inspired by the image-to-image translation model called Pix2Pix, which consists of U-Net as the generator and PatchGAN as the discriminator (Isola et al., 2017). In 10-min lead time precipitation prediction, at the Soyang-gang Dam Basin, our model outperformed the other reference models. Additionally, when we applied the recursive process to predict precipitation with lead times of up to 90 min, our model achieved adequate performance (>1/e) for R with lead times of up to about 80 min, which was an improvement over ConvLSTM (up to 60 min) and U-Net (up to 60 min). Also, the CSI and FSS (at the intensity of 0.1 mm h$^{-1}$) results for the entire domain revealed that compared to the reference models, our model generated precipitation prediction more accurately at the overall lead times. However, in the case of higher rainfall intensity, CSI and FSS showed that Rad-cGAN had relatively poor performance compared to the reference models (excluding U-net). Although our model tends to underestimate strong precipitation, the qualitative evaluation of the Typhoon Soulik confirmed that our model can capture spatiotemporal change in the area of precipitation closest to the ground truth. In addition, based on the cGAN approach, our model can generate sharper and realistic images than U-net by PSD. Furthermore, considering the edge effect, pySTEPS showed an improved performance using an extended input domain compared with the original input domain. However, because our model also performed better using extended data, we conclude that the Rad-

cGAN model is the most advanced precipitation nowcasting model that does not consider edge effects when compared to other reference models.

To develop the precipitation nowcasting model for different dam basins (Andong and Chungju Dam basins), we proposed different transfer learning strategies by using the previously trained model. Comparing the case of using transfer learning (Case 2-5) and the case of not using transfer learning (Case 1), the case of using transfer learning generally showed better performance in both Andong Dam and Chungju Dam. From the results of Cases 2 and 3 in which the performance is somewhat poor in the case of using fine-tuning, it is necessary to pay attention to the overfitting when applying the fine-tuning procedure. In addition,

when a model trained with an additional dam basins was used as a pre-trained model (Cases 4 and 5), the prediction performance was outperformed, especially at high rainfall intensity, and it was found that data diversity had an effect on model generalization and underestimation.

We confirmed that the proposed precipitation nowcasting model demonstrated improved performance over conventional machine learning-based models (U-Net and ConvLSTM) and showed that transfer learning strategies could be effectively

applied to develop models for other dam domains with summer precipitation in South Korea. However, there are remaining issues that must be considered to ensure auditability of our model for real problems, such as predicting heavy precipitation events and flash flood forecasting. First, the tendency of the model to underestimate precipitation is a major issue. The decisive cause of this issue is data imbalance, as mentioned in general machine-learning tasks (Wang et al., 2016). To address this issue, further studies need to be conducted to improve the predictive performance of extreme precipitation events by extending the

duration of training data and assigning weights to the extreme or other events. Additionally, adding information about domain characteristics, such as the digital elevation model and land cover map is expected to improve the precipitation nowcasting model. Another issue is that we trained models for different domains using basic transfer learning strategies, and evaluated the performance only for the new domains, which are not sufficient to develop models for multiple dam domains that can be used in early warning systems. To overcome this issue, for example, Wang et al. (2020) presented a new transfer learning approach

that simultaneously mined the knowledge of multiple pre-trained generators. Therefore, it is expected that further research using more advanced transfer learning strategies will help to develop a precipitation nowcasting model with good performance in all different domains to enhance practicality.

**Code and data availability**

Source code of the model architecture is available at the GitHub repository https://github.com/SuyeonC/Rad-cGAN (last

access: 13 June 2022). The pre-trained model for Soyang-gang Dam Basin and example data are available at

https://doi.org/10.5281/zenodo.6460012. The radar reflectivity composite data samples provided by the Korea

Meteorological Administration (KMA) are available at the public service:

https://data.kma.go.kr/resources/html/en/ncdci.html (last access: 11 November 2021). The dataset for the entire period can be

obtained through a separate request to the KMA.

**Author contributions**

SC and YK designed the study, and SC performed the model development, simulations, and result analysis under the

supervision of YK. SC wrote the original manuscript, and YK reviewed and edited the manuscript.

**Competing interests**

The authors declare that they have no conflict of interest.

**Acknowledgements**

This study was supported by the Basic Science Research Program through the National Research Foundation of Korea,

which was funded by the Ministry of Science, ICT & Future Planning (No. 2020R1A2C2007670), and the Technology

Advancement Research Program through the Korea Agency for Infrastructure Technology Advancement (KAIA) grant

funded by the Ministry of Land, Infrastructure and Transport (Grant 21CTAP-C163541-01).

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

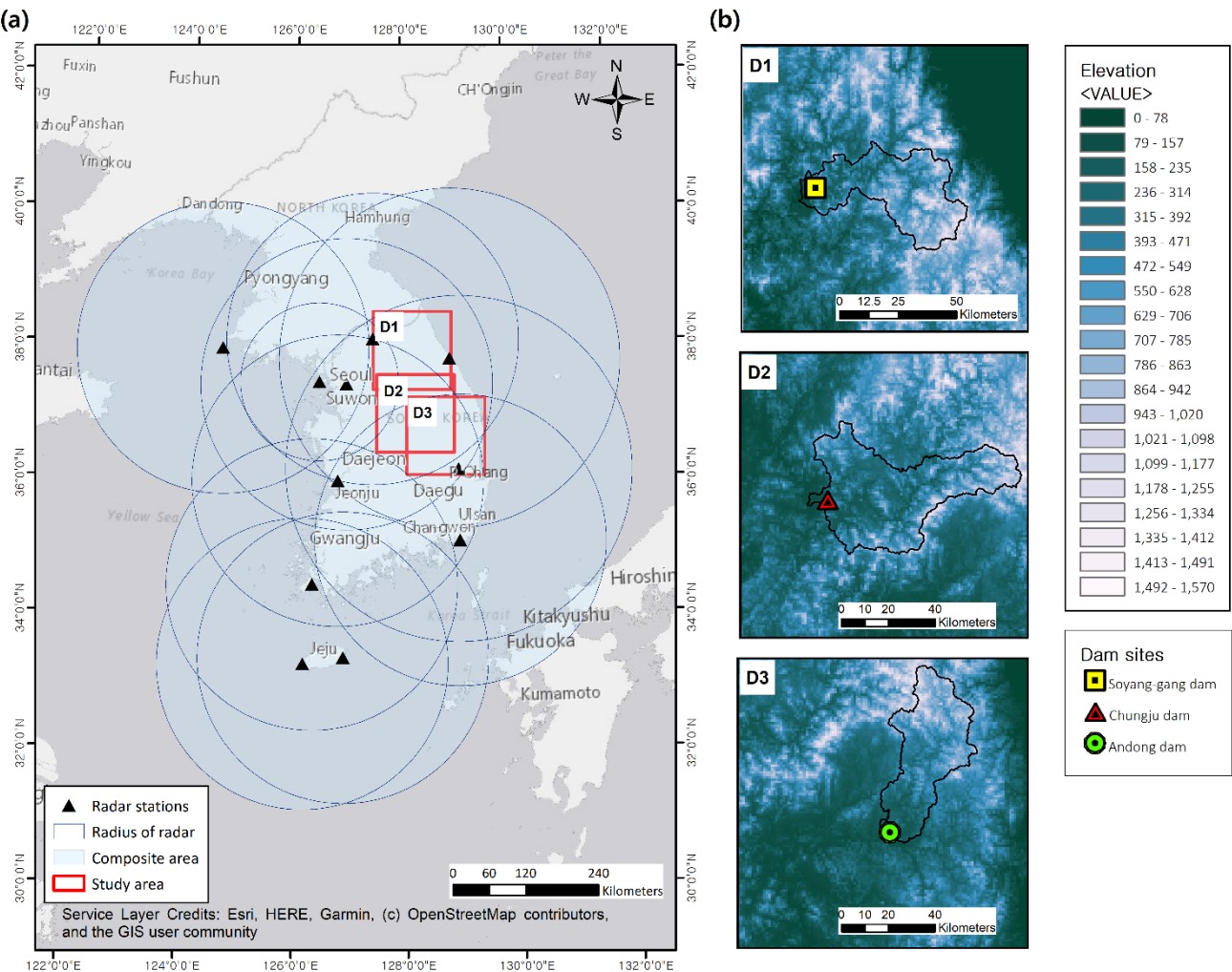

**Figure 1: (a) Composite map of radar reflectivity and location of the dam basins; (b) selected areas over the dam basin. D1, D2, and D3 represent the areas of Soyang-gang, Chungju, and Andong Dam Basins, respectively. Maps were created using ArcGIS software by Esri; Base-map source: Esri, HERE, Garmin, © OpenStreetMap contributors, and the GIS User Community.**


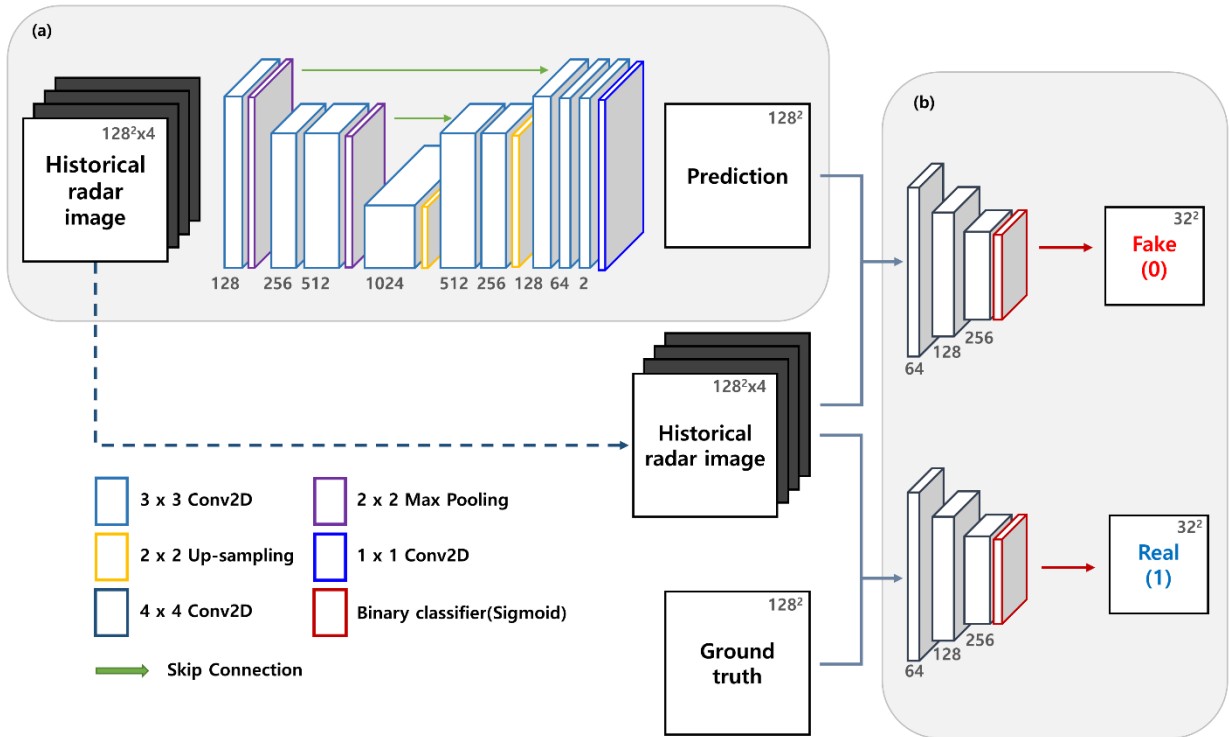

**Figure 2: Model architecture consists of a (a) generative and (b) discriminator.**

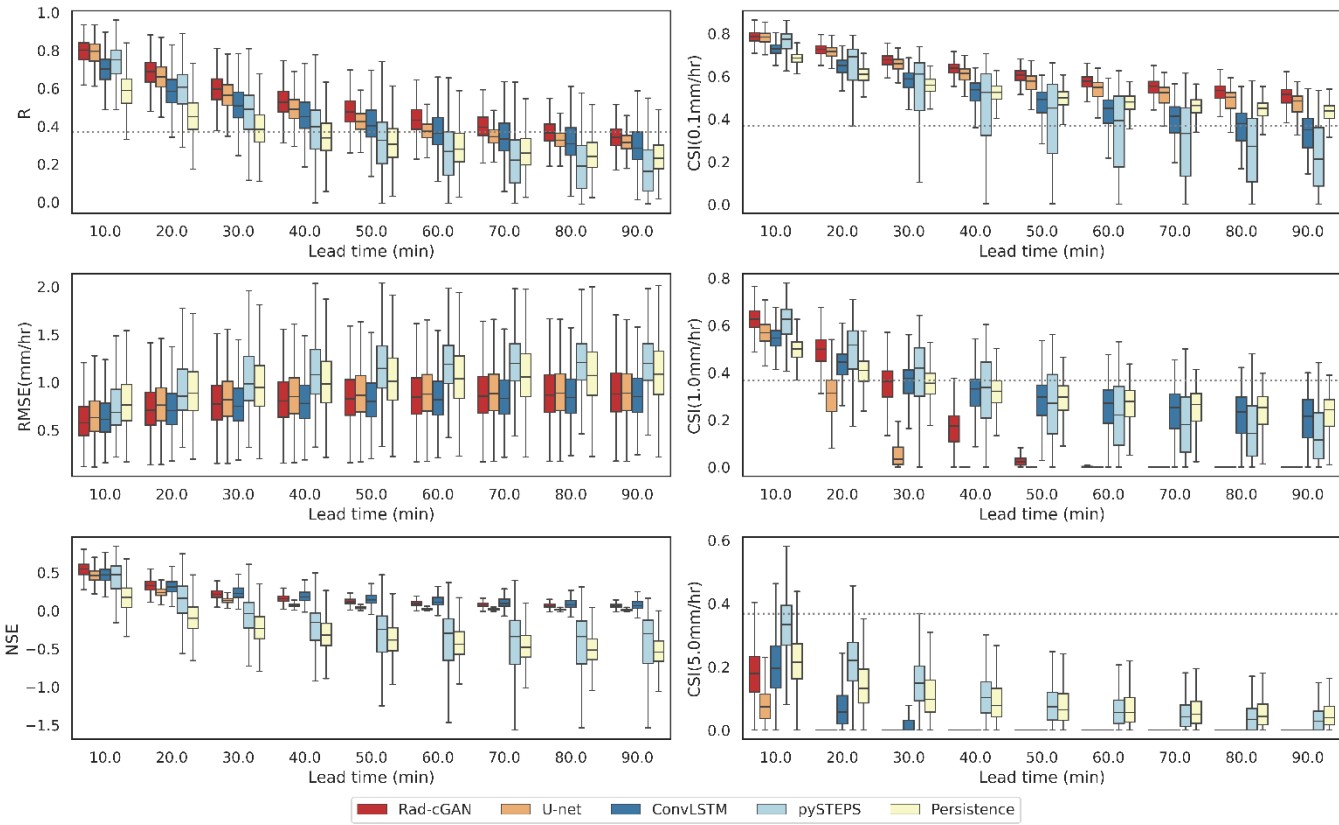


**Figure 3: Box plot of the verification metrics of model predictions at the lead time up to 90 min over all grid cells from the Soyang-gang Dam region. From top to bottom, left panels represent R, RMSE, and NSE, and right panels represent the CSI at intensity thresholds of 0.1, 1.0, and 5.0 mm h-1. Grey dotted line represents the predictability threshold (1/e≒0.37).**


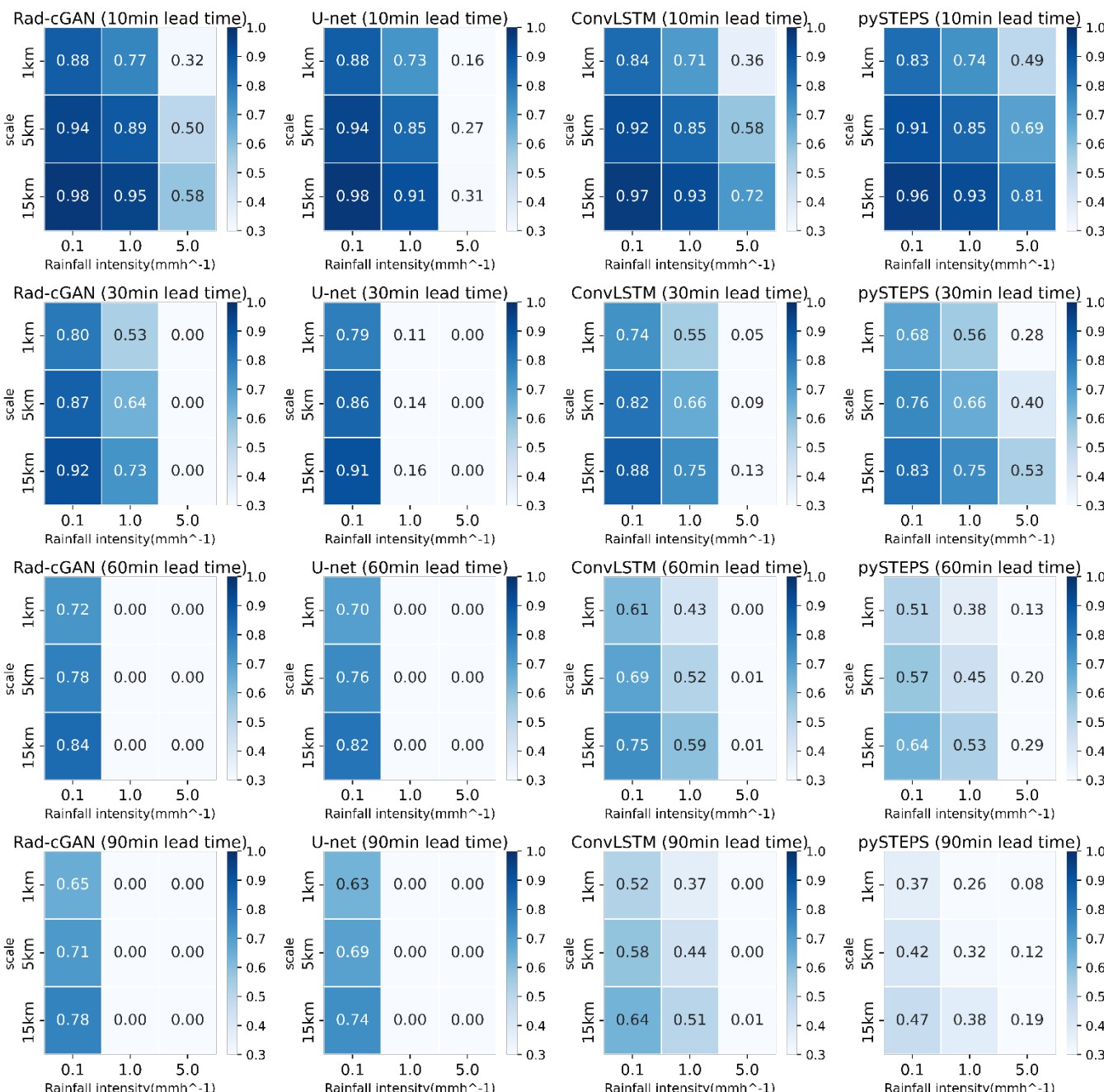

**Figure 4: Fraction Skill Scores (FSS) of model predictions at lead time of 10, 30, and 60 min at Soyang-gang Dam Basin. Panels from left to right express FSS of Rad-cGAN, U-net, ConvLSTM, and pySTEPS (S-PROG).**


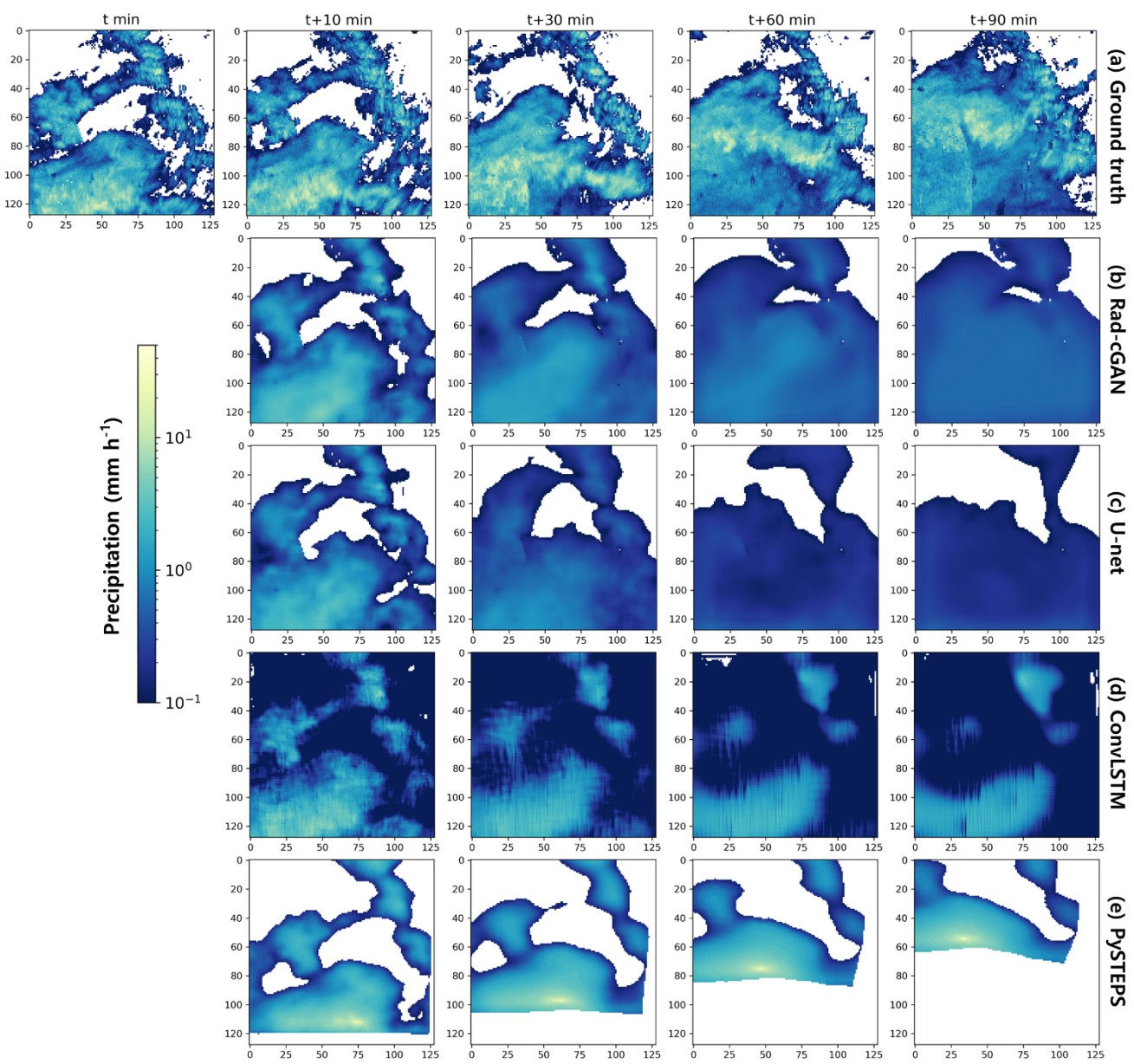

**Figure 5: Example of precipitation at forecasting time t = 23 August 2018, 17:50 UTC, for model predictions and (a) ground truth**

 **(OBS). Panels from top to bottom express ground truth: (b) prediction of Rad-cGAN model, (c) prediction of U-net based model, (d) prediction of ConvLSTM, and (e) prediction of pySTEPS.**

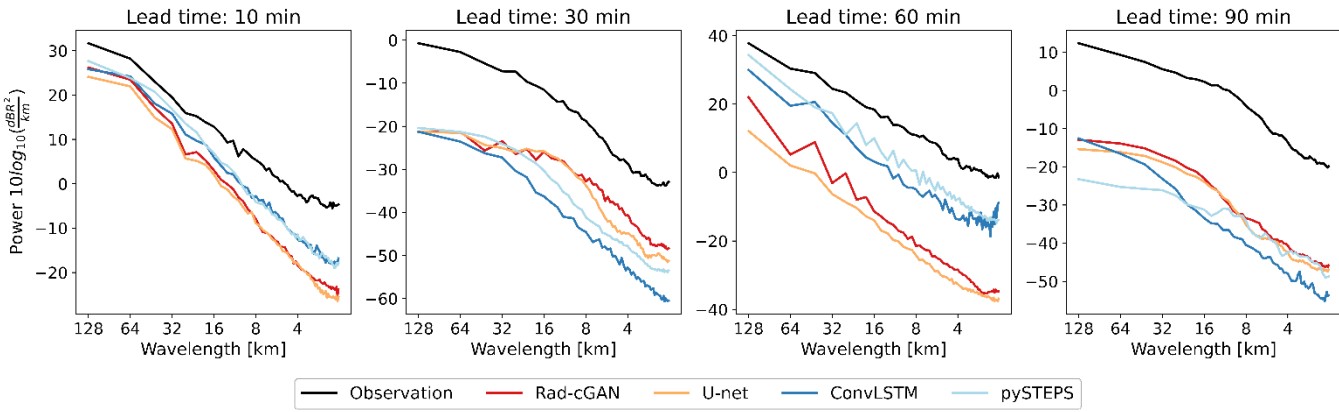

**Figure 6: Radially averaged power spectral density (PSD) at forecasting time t = 23 August 2018, 17:50 UTC, for model predictions and observation**

.

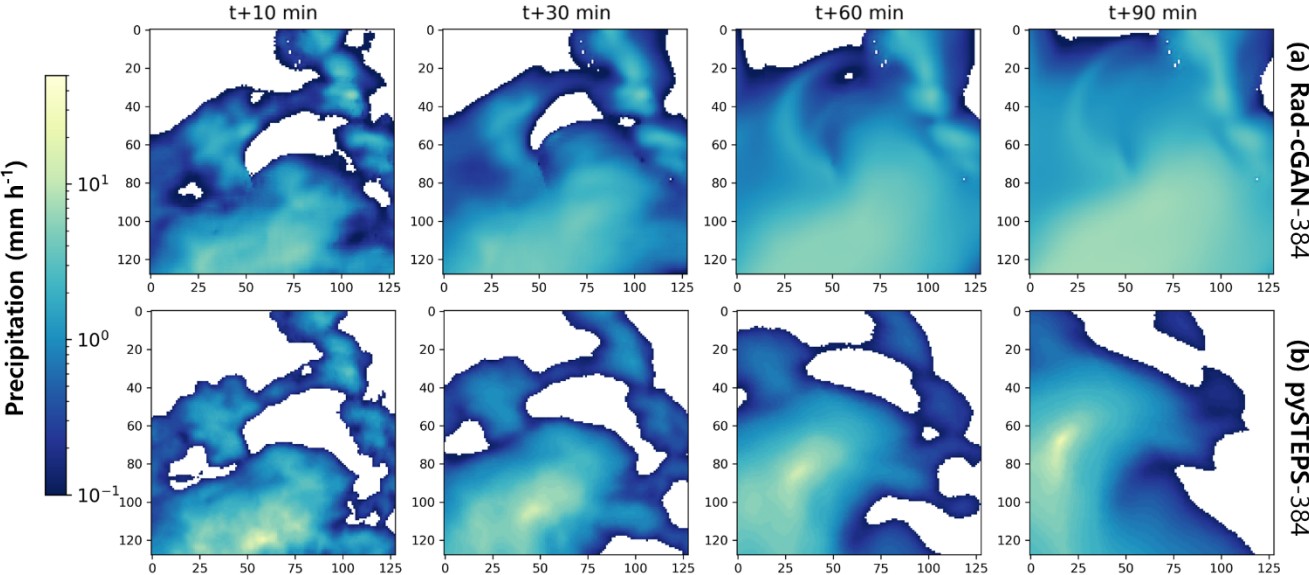

**Figure 7: Example of precipitation at forecasting time t = 23 August 2018, 17:50 UTC, for model prediction using increased input**

**area (384 × 384). Panels from top to bottom express (a) prediction of Rad-cGAN model, and (e) prediction of pySTEPS.**

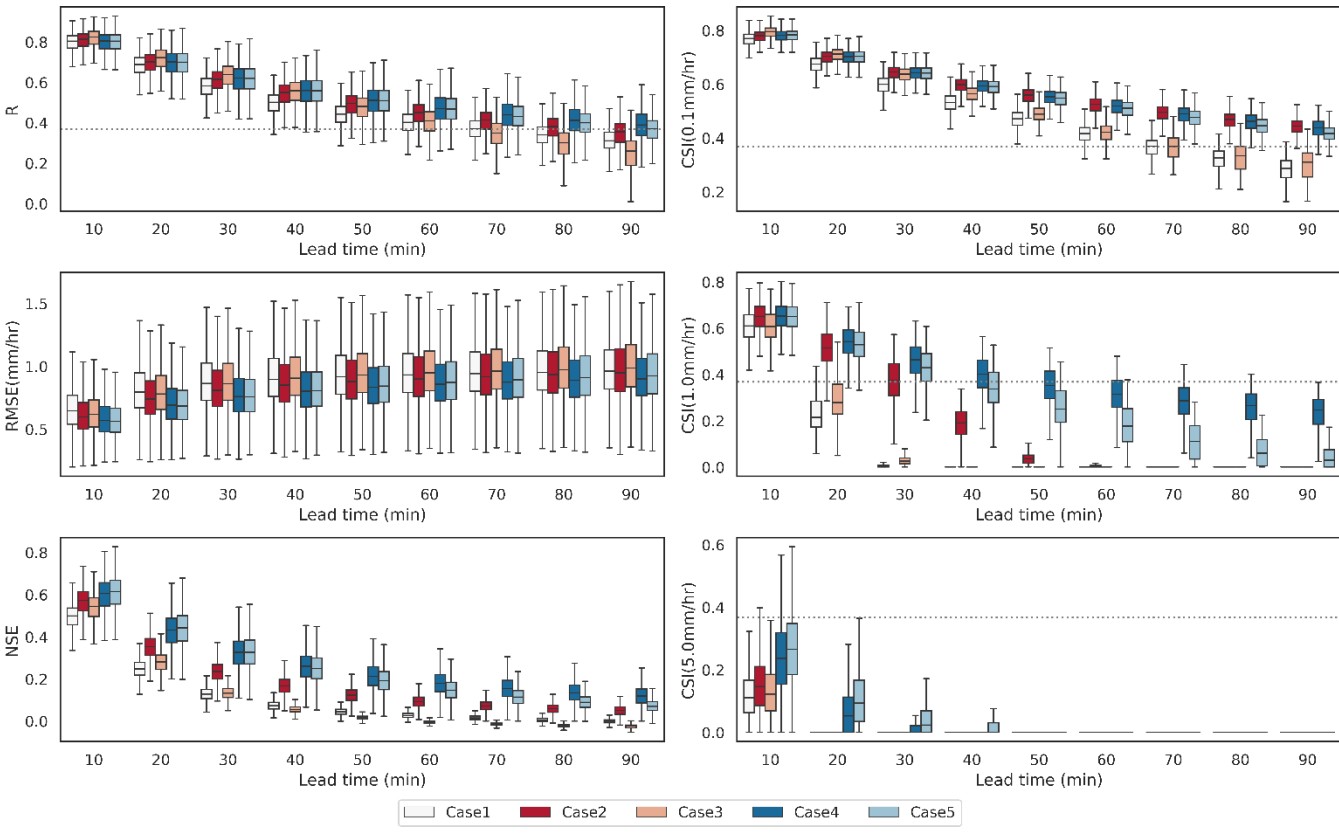

**Figure 8: Box plot of the verification metrics of model predictions at lead time up to 90 min over all grid cells from the Andong Dam Basin. (a) R, (b) NSE, (c) RMSE, and (d, e, f) CSI at intensity thresholds of 0.1, 1.0, and 5.0 mm h-1, respectively. Grey dotted line represents the predictability threshold (1/e≒0.37).**


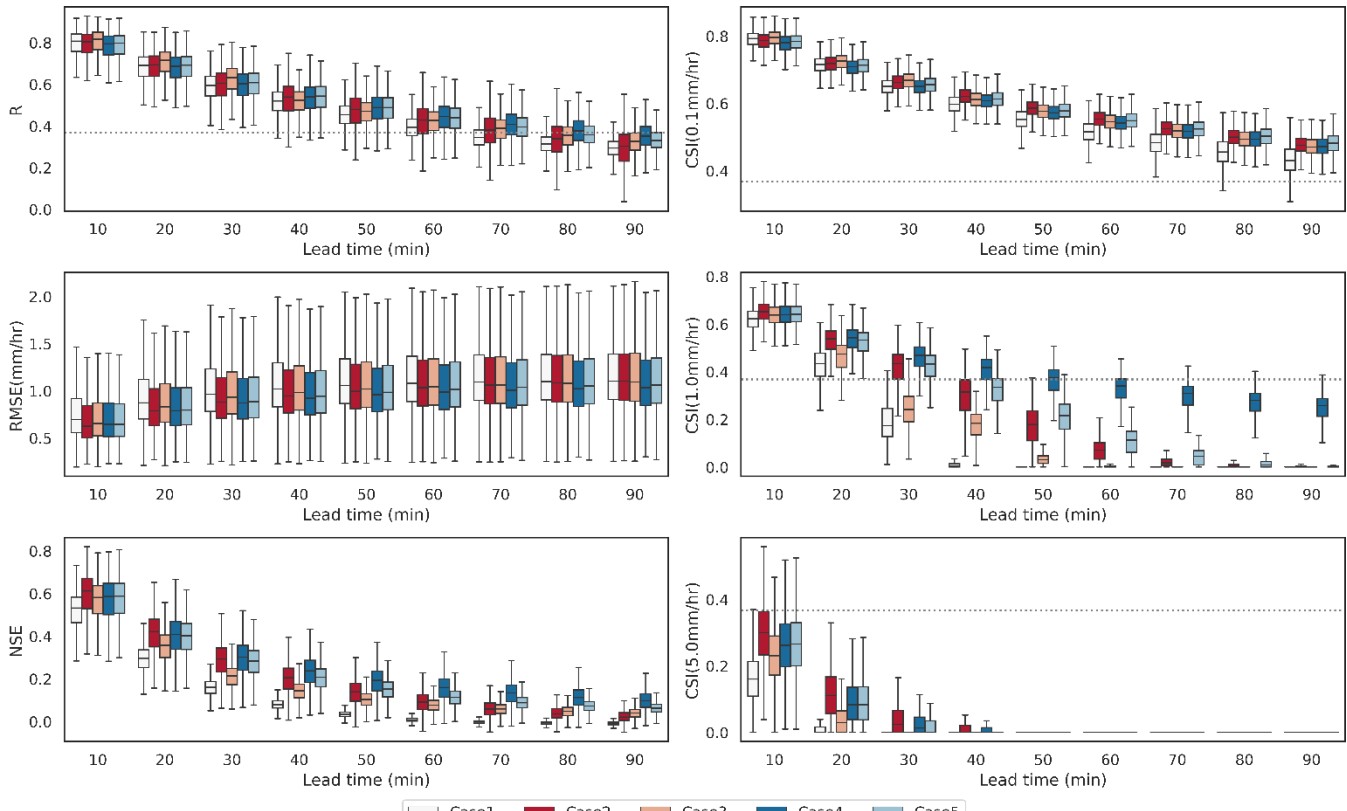

**Figure 9: Box plot of the verification metrics of model predictions at the lead time up to 90 min over all grid cells from the Chungju Dam Basin. (a) R, (b) NSE, (c) RMSE, and (d, e, f) CSI at intensity thresholds of 0.1, 1.0, and 5.0 mm h-1, respectively. Grey dotted line represents the predictability threshold (1/e≒0.37).**

**Table 1: Distribution of precipitation amount and number of examples of (a) train dataset, and (b) test dataset of each dam domains and sites.**

**(a) Train dataset**

| Interval in mm h-1 | Soyang-gang Dam Basin | Soyang-gang Dam site | Andong Dam Basin | Andong Dam site | Chungju Dam Basin | Chungju Dam site |
|---|---|---|---|---|---|---|
| 0 ≤ R < 0.1 | 84.63 | 85.00 | 83.04 | 83.57 | 84.81 | 85.03 |
| 0.1 ≤ R < 1.0 | 10.23 | 10.19 | 11.56 | 11.02 | 10.75 | 10.61 |
| 1.0 ≤ R < 4.0 | 3.78 | 3.35 | 3.91 | 3.89 | 3.43 | 3.30 |
| 4.0 ≤ R < 8.0 | 0.92 | 1.03 | 0.94 | 0.98 | 0.68 | 0.69 |
| 8.0 ≤ R < 10.0 | 0.18 | 0.21 | 0.20 | 0.23 | 0.13 | 0.16 |
| 10.0 ≤ R | 0.27 | 0.23 | 0.35 | 0.32 | 0.20 | 0.21 |
| No. of examples | 27,905 examples | | 29,136 examples | | 29,691 examples | |

**(b) Test dataset**

| Interval in mm h-1 | Soyang-gang Dam Basin | Soyang-gang Dam site | Andong Dam Basin | Andong Dam site | Chungju Dam Basin | Chungju Dam site |
|---|---|---|---|---|---|---|
| 0 ≤ R < 0.1 | 90.77 | 91.92 | 87.41 | 87.73 | 86.54 | 87.61 |
| 0.1 ≤ R < 1.0 | 5.77 | 5.16 | 7.59 | 7.48 | 8.63 | 8.06 |
| 1.0 ≤ R < 4.0 | 2.65 | 2.26 | 3.77 | 3.92 | 3.80 | 3.58 |
| 4.0 ≤ R < 8.0 | 0.58 | 0.42 | 0.87 | 0.54 | 0.76 | 0.51 |
| 8.0 ≤ R < 10.0 | 0.10 | 0.06 | 0.15 | 0.09 | 0.12 | 0.13 |
| 10.0 ≤ R | 0.14 | 0.17 | 0.21 | 0.24 | 0.14 | 0.11 |
| No. of examples | 9,753 examples | | 6,598 examples | | 6,137 examples | |

**Table 2: Contingency table for the categorical scores.**

| | | Observation | |
|---|---|---|---|
| | | Event detected | Event not detected |
| Prediction | Event detected | Hit | False alarm |
| | Event not detected | Miss | Correct non-event |


**Table 3: Experimental design for transfer learning strategies to train model with different domain. (a) Detailed training procedure of each strategy, (b) data used to train the model according to each strategy.**

| (a) Training strategies | | |
|---|---|---|
| No. | Generator | Discriminator |
| Case 1 | Train from the scratch | Train from the scratch |
| Case 2 | Use pre-trained parameters for one domain | - |
| Case 3 | Fine tuning pre-trained parameters for one domain[*] | Train from the scratch |
| Case 4 | Use pre-trained parameters for multiple domains | - |
| Case 5 | Fine tuning pre-trained parameters for multiple domains[*] | Train from the scratch |

| (b) Training dataset | | | |
|---|---|---|---|
| | Pre-trained domain | Andong Dam domain | Chungju Dam domain |
| Case 1 | - | 2014–2017 (JJA) at Andong Dam domain | 2014–2017 (JJA) at Chungju Dam domain |
| Case 2 | 2014-2017 (JJA) at Soyang-gang Dam domain | - | - |
| Case 3 | 2014-2017 (JJA) at Soyang-gang Dam domain | 2014–2017 (JJA) at Andong Dam domain | 2014–2017 (JJA) at Chungju Dam domain |
| Case 4 | 2014-2017 (JJA) at Soyang-gang/Daecheong/Juam/Yongdam Dam domain | - | - |
| Case 5 | 2014-2017 (JJA) at Soyang-gang/Daecheong/Juam/Yongdam Dam domain | 2014–2017 (JJA) at Andong Dam domain | 2014–2017 (JJA) at Chungju Dam domain |
| * Use 1/10[th] of original learning rate | | | |

**Table 4: Comparison of the average values of the verification metrics for a 10-min precipitation prediction of different models at the Soyang-gang Dam Basin during summer (June–August), 2018.**

| | R | RMSE (mm h$^{-1}$) | NSE | CSI (0.1 mm h$^{-1}$) | CSI (1.0 mm h$^{-1}$) | CSI (5.0 mm h$^{-1}$) |
|---|---|---|---|---|---|---|
| **Rad-cGAN** | **0.7891** | **0.6138** | **0.5367** | **0.7827** | **0.6262** | 0.1772 |
| U-Net | 0.7822 | 0.6626 | 0.4582 | 0.7783 | 0.5688 | 0.0793 |
| ConvLSTM | 0.6976 | 0.6508 | 0.4694 | 0.7247 | 0.5462 | 0.2019 |
| pySTEPS (baseline) | 0.7076 | 0.7826 | 0.4100 | 0.7181 | 0.5803 | **0.3214** |
| Persistence (baseline) | 0.5839 | 0.8117 | 0.1678 | 0.6821 | 0.4987 | 0.2197 |


**Table 5: Comparison of the average values of the verification metrics for a 10-min precipitation prediction of the five different models using different transfer learning strategies for the (a) Andong and (b) Chungju Dam Basins in the summer (June–August) of 2018.**

**(a) Andong Dam domain**

| | R | RMSE (mm h$^{-1}$) | NSE | CSI (0.1 mm h$^{-1}$) | CSI (1.0 mm h$^{-1}$) | CSI (5.0 mm h$^{-1}$) |
|---|---|---|---|---|---|---|
| Case 1 | 0.7945 | 0.8169 | 0.4926 | 0.7662 | 0.6073 | 0.1193 |
| Case 2 | 0.8037 | 0.7673 | 0.5624 | 0.7756 | 0.6482 | 0.1523 |
| Case 3 | 0.8146 | 0.7858 | 0.5351 | 0.7916 | 0.6067 | 0.1317 |
| Case 4 | 0.7952 | 0.7407 | 0.5948 | 0.7782 | 0.6497 | 0.2399 |
| Case 5 | 0.7952 | 0.7319 | 0.6051 | 0.7794 | 0.6472 | 0.2682 |

**(b) Chungju Dam domain**

| | R | RMSE (mm h$^{-1}$) | NSE | CSI (0.1 mm h$^{-1}$) | CSI (1.0 mm h$^{-1}$) | CSI (5.0 mm h$^{-1}$) |
|---|---|---|---|---|---|---|
| Case 1 | 0.7909 | 0.9221 | 0.5161 | 0.7893 | 0.6169 | 0.1639 |
| Case 2 | 0.7863 | 0.8609 | 0.5876 | 0.7831 | 0.6492 | 0.2981 |
| Case 3 | 0.7995 | 0.8849 | 0.5623 | 0.7920 | 0.6351 | 0.2324 |
| Case 4 | 0.7776 | 0.8808 | 0.5661 | 0.7761 | 0.6380 | 0.2614 |
| Case 5 | 0.7809 | 0.8783 | 0.5685 | 0.7803 | 0.6386 | 0.2657 |