# Peer review of "Rad-cGAN v1.0: Radar-based precipitation nowcasting model with conditional Generative Adversarial Networks for multiple dam domains"

_Geoscientific Model Development, 2021_

## Referee Comment (RC1)

**Referee report on Choi and Kim (2021), submitted to GMD**

**Subject and scope**

In their manuscript *"Rad-cGAN v1.0: Radar-based precipitation nowcasting model with conditional Generative Adversarial Networks for multiple domains"*, S. Choi and Y. Kim evaluate the performance of different deep learning designs for precipitation nowcasting. Specifically, they compare a conditional generative adversarial network (cGAN) to two previously published designs, namely ConvLSTM and U-Net. Rad-cGAN combines a U-Net design (as a generative model) with the PatchGAN design from the Pix2Pix model (as a discriminative model). Verification is carried out on a subdomain of the Korean national radar composite. Furthermore, the transferability of the trained network to other subdomains is evaluated based on different transfer learning techniques.

**Overall evaluation**

The exploration of deep learning architectures for precipitation nowcasting is gaining more and more attraction. Several studies have been published in recent years which not only suggested various network designs, but also revealed typical weaknesses and challenges of machine learning techniques, as compared to well-established heuristic techniques based on tracking and extrapolation. Further exploration is certainly warranted. Specifically, I very much welcome the investigation of transferability, as included in this study.

But as much as we need such studies, I have several major concerns that should be addressed before this manuscript is considered for publication. This would require some fundamental changes and enhancements of the analysis, hence I recommend major revisions. I will elaborate my concerns in the following.

**Major comments**

**Context with Ravuri et al. (2021)**

In their study, Choi & Kim suggest a conditional Generative Adversarial Network. A similar approach had been suggested by Ravuri et al. (2021) with good success for the UK. Unfortunately, Choi and Kim do not put their own design in context with the work of Ravuri et al. (2021). It would be helpful to point out, justify and discuss differences in the network design, and resulting implications.

**Spatial verification set-up**

The study is about the development and evaluation of different deep learning designs for precipitation nowcasting in Korea. Surprisingly for me, Choi & Kim limit the model verification to arbitrary spatial subsets of their model domain: first and foremost, to the location of the Soyang-gang dam, and second, to the upstream catchment of the dam. While I cannot see

any hydrological justification to predict the precipitation at the dam location itself, I can understand, in the context of dam operation and early warning, the relevance of predicting precipitation for the dam's catchment. However, as the paper is about nowcasting methods, the limitation to the catchment area is unwarranted as it unnecessarily reduces the amount of data that is available for verification. Besides, I do not understand why the authors first compute the verification metrics for each pixel in the catchment separately and then compute the metric's median from this. In my view, using the median improves the resulting metrics specifically for the Rad-cGAN model since it is prone to produce outliers, as the authors state themselves.

**Lack of a competitive benchmark**

It has become - and rightly so - the standard in deep learning benchmarking studies to use at least one competitive benchmark model in order to demonstrate the added value of the data-driven models. Several Python libraries have become available in recent years which allow to generate strong benchmark predictions based on tracking (e.g. optical flow) and extrapolation. The authors themselves cite e.g. PySTEPS (Pulkkinen et al., 2019) and rainymotion (Ayzel et al., 2019). I would like to ask the authors to include at least one strong (and open) benchmark from any such library.

**Insufficient metrics**

One of the major issues of deep learning models for precipitation nowcasting is that they struggle to predict intense precipitation, and that they introduce spatial smoothing to account for predictive uncertainty. The smoothing effect becomes increasingly pronounced over lead lead time if the model is applied recursively. This important issue needs to be explicitly and extensively addressed in the present study, specifically since Ravuri et al. (2021) appeared to have made substantial progress on that matter. To this end, various de-facto standards have emerged, e.g. to provide skill scores (such as the CSI) for higher intensity thresholds (up to *at least* 10 mm/h), to evaluate the power spectral density (PSD) for various lead times, and to show the fractions skill score over various spatial scales, lead times and intensity thresholds. The use of correlation, RMSE, NSE and CSI for a threshold of only 0.1 mm/h does not meet these standards.

**Transfer learning, hyperparameters**

In my opinion, the transfer learning experiment is the most interesting part of this study, yet it requires further attention and analysis. This includes the following aspects:

- In ll. 361 ff., the authors *"[...] infer that by using transfer learning, a model can be successfully developed with different domains, although it does not optimize the hyperparameter to fit the model with the new domain."* The issue of hyperparameter tuning was not addressed in the manuscript before, though. Which hyperparameters were tuned for the Soyang-gang domain, and to which effect? What are the implications for evaluating the transfer learning if you do not analyse the effects of hyperparameter tuning?

● Of course, case 1 provides an important reference for evaluating cases 2 and 3: What does it mean if cases 2 or 3 outperform case 1 in which the model is fully retrained? In addition, I recommend adding another case: the evaluation of the model without any transfer learning, just using the pretrained weights. I think this is important to appreciate the effects of transfer learning.

**Model software and reproducibility**

For model description papers, GMD states that *"[...] code must be published on a persistent public archive with a unique identifier for the exact model version described in the paper or uploaded to the supplement"*. Some code is available on a GitHub repository (https://github.com/SuyeonC/Rad-cGAN), yet, this does neither qualify as a persistent public archive nor as a unique identifier. Instead, the published model version requires a persistent DOI. Furthermore, I am not satisfied with the level of reproducibility provided by the GitHub repository: it lacks sufficient documentation (or, strictly speaking, it is not documented *at all*), it lacks the benchmark model implementations (U-Net, ConvLSTM), and it lacks a minimal working example with corresponding data and pre-trained weights. Speaking of data, I could not find any way to download the radar reflectivity composite data samples as pointed out by the authors in the "Code and data availability" section with the provided URL (https://data.kma.go.kr/resources/html/en/ncdci.html). Maybe this works in the Korean version of the website, but this is not sufficient for a study to be published in GMD. Instead, I suggest that the data (or at least samples) are included in another persistent, openly available repository, and that the authors provide sufficient guidance and a working example how to use the data with their code.

**Presentation quality**

The presentation quality of the manuscript needs to be improved. This particularly applies to the quality of the figures which all have a very low resolution. For rainfall maps, make sure that you use colormaps that are appropriate for colour-blind people.

**Specific comments**

- I think that two statements in the introduction are incorrect: NWP are *not* the standard tool for nowcasting (l. 30), and rainymotion is *not* data-driven (as suggested in l. 35)

- Formatting of equations is odd: it is difficult to separate them from the main text due to the lack of vertical spacing.

- ll. 82 ff: Speculation - topography does not necessarily suggest anything on rainfall patterns (whatever is meant by "rainfall patterns").

- Fig.2: Please add spatial dimensions to the presented data volumes. Furthermore, for the discriminative model (subplot b), it is not clear how PatchGAN's output (34x34) compares to ground truth (pixelwise, averaging, etc.).

- Model description and analysis:

- The authors stated that the size of the optimised patch is 34x34. However, it is not clear how that patch is clipped from the generated/ground truth image – the output of the discriminative model has a spatial dimension of 32x32 suggesting that there is some overlapping strategy.

- ll.141-142 state *"...each pixel of the output referred to the probability that the discriminative model determines each patch of the input pair as the real one."* It would be interesting to see the corresponding results on some real examples in the analysis.

- Based on Goodfellow (NIPS 2016 Tutorial): *"...If both models have sufficient capacity, then the Nash equilibrium of this game corresponds to the G(z) being drawn from the same distribution as the training data, and D(x) = 1/2 for all x."* It would be interesting to see the corresponding results for the discriminative model on some real examples.

- ll. 227 ff.: "Since the precipitation prediction of the model was more accurate, the prediction and observation showed a strong positive linear relationship." - doesn't make sense to me.

- Fig. 6: In order to appreciate the spatial patterns, I would prefer to see the predicted rainfall instead of the bias. If bias plays a role, it should be expressed in adequate verification metrics.

- I don't think that such metrics should be presented with a precision of 4 digits.

- Fig. 3: I do not see the added value of the (right hand) time series panel - there is not much to see and learn when you look at two months of 10 min data.

- I don't see the need for Tab. 3 when you have Fig. 4.

- L. 277: "All models except for persistence performed extremely well" - what is the basis for such a strong statement?

- L. 283-285: "our model performs better than U-Net in predicting peak precipitation [...] prediction accuracy of the ConvLSTM model for maximum precipitation was higher than that of our model" - please confirm these statements by adequate metrics, not from visual inspection

- Ll. 287 ff: what is the basis for normative statements such as "good" and "sufficient"?

- Why use an entirely different presentation format in Fig. 5 to evaluate the performance in the catchments? Anyway, a revised version of the paper should not evaluate model performance for the dam locations and the catchments.

---

## Author Comment (AC1)

**[RC1]**

We thank the reviewers for their constructive comments on our manuscript. In the following paragraphs, the reviewers' comments are in black font, and our point-by-point responses are in blue.

**Major comments**

**Context with Ravuri et al. (2021)**

In their study, Choi & Kim suggest a conditional Generative Adversarial Network. A similar approach had been suggested by Ravuri et al. (2021) with good success for the UK. Unfortunately, Choi and Kim do not put their own design in context with the work of Ravuri et al. (2021). It would be helpful to point out, justify and discuss differences in the network design, and resulting implications.

➔ As pointed out by the reviewer, we have added following paragraph about differences between our work and the work of Ravuri et al. (2021) to clarify our intentions.

L74: *"The aim of the present study was to develop an advanced precipitation nowcasting model for multiple dam basins that can be applied as an early warning system. The decision-making process at upstream dams with regard to flood control, which is directly related to urban and rural water management, influences flood risk considerably. From such a dam management perspective, water level and precipitation prediction at dam sites are major factors to be considered, suggesting that increasing rainfall prediction accuracy at the dam site is essential for effective flood management. To develop an advanced precipitation nowcasting model with good prediction performance for dam sites and basins in general, we designed a model based on the cGAN approach (Rad-cGAN) for multiple dam domains of the Soyang-gang, Andong, and Chungju dam basins in South Korea."*

L127: *"In this study, we developed a radar-based precipitation nowcasting model using a cGAN framework. Recently, research on weather prediction using cGAN, an advanced machine learning approach, has been conducted extensively (e.g., Rüttgers et al., 2019; Ravuri et al., 2021). For example, Ravuri et al. (2021) proposed a generator consisting of two modules; conditioning stack (using CNN to extract representation of input); and sampler (using ConvGRU to generate prediction). The model, which used ConvGRU, could observe spatiotemporal changes of inputs such as ConvLSTM, and attempted to improve performance by extracting features from different spatial*

*dimensions and deriving the results. Whereas the generator used to predict future radar map, the discriminator used a dual architecture that distinguishes the real and generated frames, to ensure both temporal and spatial consistency. Unlike the model proposed by Ravuri et al. (2021), our model adopts a U-net architecture that uses a CNN layer in image generation based on the underlying Pix2Pix model; the architecture exhibits outstanding performance in image-to-image translation tasks (Isola et al., 2017). Also, we considered only spatial consistency in the PatchGAN discriminator, which distinguishes images for each N × N patch (N can be smaller than the full size of the image). U-net-based precipitation nowcasting model has previously demonstrated performance superior to that of a traditional radar-based precipitation nowcasting model that uses optical flow (Ayzel et al., 2020). Therefore, here, we apply the basic cGAN methodology to the U-net structure to improve performance and confirm the applicability of the transfer learning methodology to multiple domains.*"

**Spatial verification set-up**

The study is about the development and evaluation of different deep learning designs for precipitation nowcasting in Korea. Surprisingly for me, Choi & Kim limit the model verification to arbitrary spatial subsets of their model domain: first and foremost, to the location of the Soyang-gang dam, and second, to the upstream catchment of the dam. While I cannot see any hydrological justification to predict the precipitation at the dam location itself, I can understand, in the context of dam operation and early warning, the relevance of predicting precipitation for the dam's catchment. However, as the paper is about nowcasting methods, the limitation to the catchment area is unwarranted as it unnecessarily reduces the amount of data that is available for verification. Besides, I do not understand why the authors first compute the verification metrics for each pixel in the catchment separately and then compute the metric's median from this. In my view, using the median improves the resulting metrics specifically for the Rad-cGAN model since it is prone to produce outliers, as the authors state themselves.

➔ To clarify why we evaluate model for dam site and whole basin, we have added more explanations to explain our main objective, as follows:

L74: "*The aim of the present study was to develop an advanced precipitation nowcasting model for multiple dam basins that can be applied as an early warning system. The decision-making process at upstream dams with regard to flood control, which is directly related to urban and rural water*

*management, influences flood risk considerably. From such a dam management perspective, water level and precipitation prediction at dam sites are major factors to be considered, suggesting that increasing rainfall prediction accuracy at the dam site is essential for effective flood management. To develop an advanced precipitation nowcasting model with good prediction performance for dam sites and basins in general, we designed a model based on the cGAN approach (Rad-cGAN) for multiple dam domains of the Soyang-gang, Andong, and Chungju dam basins in South Korea.*"

As per the reviewer's suggestion, we have revised the resulting metrics as boxplots:

[Figure]

*Figure 5: Box plot of verification metrics of model predictions at the lead time up to 90 min over all grid cells from the Soyang-gang Dam region. Left panels from top to bottom represent R, RMSE, NSE, and right panels from top to bottom represent CSI at intensity threshold of 0.1, 1.0, 5.0 mm h-1.*

**Lack of a competitive benchmark**

It has become - and rightly so - the standard in deep learning benchmarking studies to use at least one competitive benchmark model in order to demonstrate the added value of the data-driven models. Several Python libraries have become available in recent years which allow to generate strong benchmark predictions based on tracking (e.g. optical flow) and extrapolation. The authors themselves cite e.g. PySTEPS (Pulkkinen et al., 2019) and rainymotion (Ayzel et al., 2019). I would

like to ask the authors to include at least one strong (and open) benchmark from any such library.

➡ As per reviewer's suggestion, we used PySTEPS to predict future precipitation based on our test dataset (summer of 2018), and the results have been added as a benchmark model for use in evaluation of model performance.

L211: *"2.3.1 PySTEPS*

*PySTEPS (Pulkkinen et al., 2019) is the open-source and community-driven python framework for radar-based probabilistic precipitation nowcasting that is considered a strong baseline model (Imhoff et al., 2019; Ravuri et al., 2021). In the present study, STEPS (Short-Term Ensemble Prediction System) (Bowler et al., 2006) nowcast ensemble from the pySTEPS library was used as the benchmark model. To generate precipitation predictions, we first provided input precipitation images (unit: dBR) that were transformed from four consecutive radar reflectivity images (from t-30 to t) based on a Z-R relationship (Eq. (1)). The transformed precipitation was also used to estimate motion field, and the motion field and precipitation were used as input data in the STEPS model. Future precipitation at a lead time of up to 90 min for the test period (JJA of 2018) was generated based on the average across 20 ensemble members from the results of STEPS nowcasts. The source code of pySTPES is available in GitHub (https://pysteps.github.io, last accessed: 5 April 2022)."*

**Insufficient metrics**

One of the major issues of deep learning models for precipitation nowcasting is that they struggle to predict intense precipitation, and that they introduce spatial smoothing to account for predictive uncertainty. The smoothing effect becomes increasingly pronounced over lead lead time if the model is applied recursively. This important issue needs to be explicitly and extensively addressed in the present study, specifically since Ravuri et al. (2021) appeared to have made substantial progress on that matter. To this end, various de-facto standards have emerged, e.g. to provide skill scores (such as the CSI) for higher intensity thresholds (up to at least 10 mm/h), to evaluate the power spectral density (PSD) for various lead times, and to show the fractions skill score over various spatial scales, lead times and intensity thresholds. The use of correlation, RMSE, NSE and CSI for a threshold of only 0.1 mm/h does not meet these standards.

➡ As per the reviewer's suggestion, we have added the CSI for higher rainfall intensity thresholds. Also, we have added fraction skill score (FSS) and power spectral density (PSD) to evaluate model

results in the revised manuscript. When calculating CSI and FSS, we used intensity thresholds of 0.1, 1.0, and 5.0 mm/h, and when calculating FSS, we used neighborhood sizes of 1, 5, and 15 km.

L298: *"We used the CSI (Eq. (8)), which is a measure of categorical forecast performance, to verify the model accuracy for precipitation event detection.*

$$CSI = \frac{hits}{hits + false\ alarms + misses}$$

$$(8)$$

*where* $hits$ *(correct event forecasts),* $false\ alarms$ *(incorrect event forecasts), and* $misses$ *(missed events) are defined by a contingency table (Table 2). Also, FSS can spatially verify model performance by comparing fraction of grid points of prediction and ground truth, which exceed certain rainfall intensity thresholds within the neighborhood (Eq. (9)).*

$$FSS = 1 - \frac{\sum_{i=1}^{n}(P_p - P_o)^2}{\sum_{i=1}^{n}P_p{}^2 + \sum_{i=1}^{n}(P_o)^2}$$

$$(9)$$

*where* $P_p$ *and* $P_o$ *are the fractions of prediction and observation, respectively, calculated by specific thresholds in neighborhood size. For calculating CSI and FSS, we selected several intensity thresholds, including 0.1, 1.0, and 5.0 mm h$^{-1}$, and for FSS, we used neighborhood sizes of 1, 5, and 15 km. Additionally, we calculated the radially averaged power spectral density (PSD) of predictions and observations to assess blurring effect of predicted image by models."*

L405: *"The results can also be confirmed through the FSS of each model (Fig. 6). When comparing Rad-cGAN and U-net, as lead time and rainfall intensity increase, both models decrease FSS; however, Rad-cGAN model exhibited superior performance. However, ConvLSTM had a relatively high FSS value under high rainfall intensity compared to those of the other two models."*

L423: *Figure 8 shows the PSD for each result in Fig. 7. Based on Fig. 7, all models exhibited a blurring effect compared to the ground truth. However, when comparing U-net and Rad-cGAN, Rad-cGAN has slightly lower blurring effect. This is because a sharper image can be generated when cGAN is applied to the U-net structure (Isola et al. (2017)), which shows that our model successfully applied the cGAN technique. Therefore, based on the overall verification metrics, we conclude that Rad-cGAN has the optimal prediction performance in nowcasting and prediction of spatial patterns of movement of*

precipitation.”

[Figure]

*Figure 6: Fraction Skill Scores (FSS) of model predictions at lead time of 10, 30, and 60 min at Soyang-gang Dam Basin. Panels from left to right express FSS of Rad-cGAN, U-net, ConvLSTM.*

[Figure]

*Figure 8: Radially averaged power spectral density (PSD) at forecasting time t = 23 August 2018, 17:50 UTC, for model predictions and observation*

**Transfer learning, hyperparameters**

In my opinion, the transfer learning experiment is the most interesting part of this study, yet it requires further attention and analysis. This includes the following aspects:

● In ll. 361 ff., the authors "[...] infer that by using transfer learning, a model can be successfully developed with different domains, although it does not optimize the hyperparameter to fit the model with the new domain." The issue of hyperparameter tuning was not addressed in the manuscript before, though. Which hyperparameters were tuned for the Soyang-gang domain, and to which effect? What are the implications for evaluating the transfer learning if you do not analyse the effects of hyperparameter tuning?

➔ To decide the model structure and training setup, we tuned the following hyperparameters: number of layers, number of hidden nodes, convolution filter size, patch size (defined by structure of discriminator), batch size and learning rate. We used radar 2014–2016 data (June–August) and 2017 data (June–July) to train model of each combination. Then, 2017 data (August) were used to calculate mean absolute error (MAE) and CSI (at an intensity threshold of 0.1mm/h) to obtain the optimal combination of hyperparameters. The results of each model showed an MAE range of 0.4514~41.6607 and a CSI range of 0.0~0.8322. As such, hyperparameters have a great influence on model performance, and we pointed out that if we use an already optimal model through transfer learning, sufficient performance can be obtained without any effort to tune the hyperparameters.

We have added paragraph about hyperparameter tuning in revised manuscript.

L173: "*Before proceeding with training to optimize the model for the input data, hyperparameter*

*tuning is required to determine the most optimal model structure and training settings. We selected the following hyperparameters: number of layers, number of hidden nodes, convolution filter size, patch size, batch size, and learning rate. To select the appropriate hyperparameter combination, the model for each combination was trained using radar data from 2014 to 2016 (June to August) and data from 2017 (June to July). Subsequently, using data from 2017 (August), the mean absolute error (MAE) and critical success index (CSI) (at an intensity threshold of 0.1 mm/h) were calculated to obtain the optimal combination of hyperparameters. Based on the tuning results, the MAE range was 0.45–47.66 and the CSI range was 0.0–0.83, and the results confirmed that hyperparameters influence model performance considerably. Based on the combinations that performed optimally, we determined the model structure and training settings."*

● Of course, case 1 provides an important reference for evaluating cases 2 and 3: What does it mean if cases 2 or 3 outperform case 1 in which the model is fully retrained? In addition, I recommend adding another case: the evaluation of the model without any transfer learning, just using the pretrained weights. I think this is important to appreciate the effects of transfer learning.

➔ There was a mistake in the original manuscript about the explanation of case 2 strategies. Case 2 had weights similar to those of pre-trained model; therefore, we already evaluated the model without transfer learning. This has been corrected, and a few more transfer strategies have been added to Table 2 in the revised manuscript as follow:

L333: *"We used transfer learning to train our model for different dam basins, i.e., Andong and Chungju, with a pre-trained model that was completely trained by data from the Soyang-gang Dam Basin. In addition, in existing papers that have successfully applied the transfer learning strategies, it was used to develop a model for a new domain using a pre-trained model based on vast data. Consequently, we used the pre-trained model with Daecheong Dam, Juam Dam, and Yongdam Dam basin data, in addition to Soyang-gang Dam data, to assess the amount of data required to develop a model for a new dam domain. The selected strategies were inspired by a previous approach of transferring GAN (Wang et al., 2018; Mo et al., 2020). We formulated two strategies for each pre-trained model. First, the weights of the pre-trained generator were frozen and used directly in new dam domain (Case 2, 4). Next, to the weights of the pre-trained generator were fine-tuned (1/10th of the original learning rate) and the discriminator trained (Case 3, 5). In addition, the entire model was trained for the new domain (Case 1) (Table 3a). The model was trained for the Chungju and*

*Andong Dam domains, separately, using the five strategies (Table 3b).”*

*Table 2: Experimental design for transfer learning strategies to train model with different domain. (a) Detailed training procedure of each strategy, (b) data used to train the model according to each strategy.*

| (a) Training strategies | | |
|---|---|---|
| No. | Generator | Discriminator |
| Case 1 | Train from the scratch | Train from the scratch |
| Case 2 | Use pre-trained parameters for one domain | - |
| Case 3 | Fine tuning pre-trained parameters for one domain* | Train from the scratch |
| Case 4 | Use pre-trained parameters for multiple domains | - |
| Case 5 | Fine tuning pre-trained parameters for multiple domains* | Train from the scratch |

| (b) Training dataset | | | |
|---|---|---|---|
| | Pre-trained domain | Andong Dam domain | Chungju Dam domain |
| Case 1 | - | 2014–2017 (JJA) at Andong Dam domain | 2014–2017 (JJA) at Chungju Dam domain |
| Case 2 | 2014-2017 (JJA) at Soyang-gang Dam domain | - | - |
| Case 3 | 2014-2017 (JJA) at Soyang-gang Dam domain | 2014–2017 (JJA) at Andong Dam domain | 2014–2017 (JJA) at Chungju Dam domain |
| Case 4 | 2014-2017 (JJA) at Soyang-gang/Daecheong/Juam/Yongdam Dam domain | - | - |
| Case 5 | 2014-2017 (JJA) at Soyang-gang/Daecheong/Juam/Yongdam Dam domain | 2014–2017 (JJA) at Andong Dam domain | 2014–2017 (JJA) at Chungju Dam domain |
| * Use 1/10th of original learning rate | | | |

**Model software and reproducibility**

For model description papers, GMD states that "[...] code must be published on a persistent public archive with a unique identifier for the exact model version described in the paper or uploaded to the supplement". Some code is available on a GitHub repository (https://github.com/SuyeonC/RadcGAN), yet, this does neither qualify as a persistent public archive nor as a unique identifier. Instead, the published model version requires a persistent DOI. Furthermore, I am not satisfied with the level of reproducibility provided by the GitHub repository: it lacks sufficient documentation (or, strictly speaking, it is not documented at all), it lacks the benchmark model implementations (U-Net, ConvLSTM), and it lacks a minimal working example with corresponding data and pre-trained weights. Speaking of data, I could not find any way to download the radar reflectivity composite data samples as pointed out by the authors in the "Code and data availability" section with the provided URL (https://data.kma.go.kr/resources/html/en/ncdci.html). Maybe this works in the Korean version of the website, but this is not sufficient for a study to be published in GMD. Instead, I suggest that the data (or at least samples) are included in another persistent, openly available repository, and that the authors provide sufficient guidance and a working example how to use the data with their code.

➔ As pointed out by the reviewer, we will provide more documentation and the pre-trained models (i.e., Rad-cGAN and reference models) in our GitHub repository to share our work. Also, for the data availability, as mentioned in the "Code and data availability" statement, the entire dataset can be obtained through a separate request to KMA; therefore, we will add some sample data (e.g., typhoon Soulik event) in our GitHub repository.

**Presentation quality**

The presentation quality of the manuscript needs to be improved. This particularly applies to the quality of the figures which all have a very low resolution. For rainfall maps, make sure that you use colormaps that are appropriate for colour-blind people.

**Specific comments**

- I think that two statements in the introduction are incorrect: NWP are not the standard tool for nowcasting (l. 30), and rainymotion is not data-driven (as suggested in l. 35)

➔ We have corrected it as follows:

L40: *"Among the existing precipitation nowcasting models, numerical weather prediction (NWP), which performs rainfall prediction based on atmospheric physics equations, can generate high-resolution rainfall forecasts with long lead times. However, NWP has exhibited poor forecast performance with relatively short (0–2 h) lead times (Berenguer et al., 2012)."*

L45: *"Additionally, the increased availability of high-resolution remote sensing observation data (e.g., radar) and computer resources has facilitated the development of advanced precipitation nowcasting models. For example, Ayzel et al. (2019) developed an optical flow-based precipitation nowcasting model called rainymotion…"*

- Formatting of equations is odd: it is difficult to separate them from the main text due to the lack of vertical spacing.

➔ We have corrected it.

- ll. 82 ff: Speculation - topography does not necessarily suggest anything on rainfall patterns (whatever is meant by "rainfall patterns").

➔ We have added explanations with related references in the revised manuscript, as follows:

L98: "*Since topography (especially mountainous areas such as study domains) affects atmospheric conditions, such as temperature, humidity, air pressure distribution, and cloud formation, it directly or indirectly affects rainfall formation and distribution of rainfall in an area (Basist et al., 1994; Prudhomme and Reed, 1998). Consequently, data extracted from the three domains with different topographic characteristics would exhibit different rainfall patterns.*"

Reference
*Basist, A., Bell, G. D., & Meentemeyer, V.: Statistical relationships between topography and precipitation patterns. Journal of climate, 7(9), 1305-1315, 1994.*
*Prudhomme, C. and Reed, D.W.: Relationships between extreme daily precipitation and topography in a mountainous region: a case study in Scotland. Int. J. Climatol., 18: 1439-1453, 1998.*

- Fig.2: Please add spatial dimensions to the presented data volumes. Furthermore, for the discriminative model (subplot b), it is not clear how PatchGAN's output (34x34) compares to ground truth (pixelwise, averaging, etc.).

➔ As per reviewer's suggestion, we have added spatial dimensions in Fig. 2, as follows:

[Figure]

*Figure 2. Model architecture consists of a (a) generative and (b) discriminative model.*

The output of PatchGAN size is not 34 × 34. The patch size (34) is the size of the receptive field, which is the region in the input image that is used to measure one pixel of output feature. So, the patch size was defined based on structure of discriminator (e.g., number of layers, nodes, and filter sizes). Also, while training PatchGAN, we generated the ground truth of the training dataset, which was labeled based on input dataset (if the input data is an actual image, the resulting image is filled with 1, otherwise it is filled with 0). We have added this explanation about PatchGAN in the revised manuscript to avoid any confusion.

L158: "*Particularly, PatchGAN only penalizes the structures over a certain scale of image patches; therefore, the discriminator classifies whether the N × N patch in the input pair is real or fake. This patch represents the receptive field, which is the region in the input image that is used to measure the associated feature of the output layer. Consequently, the size of the patch (N) was determined based on the structure of the entire discriminator (e.g., number of layers, nodes, filter size, paddings, and strides), and it increased as the model became deeper. We constructed a discriminator model with a 34 × 34 patch size through hyperparameter tuning.*"

- Model description and analysis:
  - The authors stated that the size of the optimised patch is 34x34. However, it is not clear how that

patch is clipped from the generated/ground truth image – the output of the discriminative model has a spatial dimension of 32x32 suggesting that there is some overlapping strategy.

➔ As in the response to the previous comment, patch size is the region used to measure the output feature (e.g., neuron of each hidden layer or each pixel value of output image), which is automatically defined by the structure of a discriminator. So, it is not the manually cropped region from the input (generated/ground truth) image. We have added following explanation about it.

L167: "*To train the discriminator as a classifier, we manually generated a training dataset consisting of an input image pair and a target image, with spatial dimensions of 32 × 32 filled with 1 (for real image pairs) or 0 (for generated image pairs). Therefore, each pixel of the output estimates the probability that the discriminator determines each patch of the input pair as the real one.*"

- ll.141-142 state "...each pixel of the output referred to the probability that the discriminative model determines each patch of the input pair as the real one." It would be interesting to see the corresponding results on some real examples in the analysis.

➔Since the GAN framework calculate and use the output of the discriminator automatically, we did not design the model to obtain the output from the discriminator separately.

- Based on Goodfellow (NIPS 2016 Tutorial): "...If both models have sufficient capacity, then the Nash equilibrium of this game corresponds to the G(z) being drawn from the same distribution as the training data, and D(x) = 1/2 for all x." It would be interesting to see the corresponding results for the discriminative model on some real examples.

➔ As mentioned in Goodfellow (NIPS 2016 Tutorial), in the ideal case, the change in the loss function during training is the same as what the reviewer said. However, due to the characteristics of the GAN model, the training process is unstable compared to those of other machine learning models, so that the actual loss functions of generator and discriminator show highly oscillating shapes. Therefore, we did not show it separately because we did not find implications in showing the loss function graph.

- ll. 227 ff.: "Since the precipitation prediction of the model was more accurate, the prediction and

observation showed a strong positive linear relationship." - doesn't make sense to me.

➔ We have corrected as follows:

L286: "*As the collinearity between actual rainfall and predicted rainfall increases, the explanatory power of the rainfall simulated by the model increases, so that the performance of the model can be illustrated by strong positive linear relationship between predictions and observations. Hence, we confirmed that the model exhibits better performance when R (Eq. (5)), calculated based on the model prediction and observation, is closer to 1.*"

- Fig. 6: In order to appreciate the spatial patterns, I would prefer to see the predicted rainfall instead of the bias. If bias plays a role, it should be expressed in adequate verification metrics.

➔ We have corrected Fig. 6, which is Fig. 7 in the revised manuscript, as follows:

[Figure]

*Figure 7: Precipitation observation example at forecasting time t = 23 August 2018, 17:50 UTC, for model predictions and (a) ground truth (OBS). Panels from top to bottom express ground truth: (b) prediction of Rad-cGAN model, (c) prediction of U-net based model, (d) prediction of ConvLSTM, and € prediction of pySTEPS*

- I don't think that such metrics should be presented with a precision of 4 digits.

➔ We have corrected it

- Fig. 3: I do not see the added value of the (right hand) time series panel - there is not much to see and learn when you look at two months of 10 min data. - I don't see the need for Tab. 3 when you

have Fig. 4.

➔ Since the model was trained to generate 10-min prediction images, it is necessary to show quantitative indicators of the results after 10 minutes when evaluating the model's performance; therefore, we created Table 3 separately from Fig. 4, to highlight performance according to the increase in lead time.

- L. 277: "All models except for persistence performed extremely well"- what is the basis for such a strong statement?

Our statements are based on verification metrics. Generally, for evaluating hydrological model, the R and NSE greater than 0.5 are acceptable (Moriasi et al, 2007). Germann and Zawadzki (2002) also suggested that the threshold of predictability is 1/e≒0.37, assuming that the CSI values follow the exponential law. Considering these criteria, we determined that the model has great performance when R >0.8, NSE >0.5, and CSI >0.5. We have elaborated our statement with references in revised manuscript.

L349: *"As the general criterion for evaluating hydrological models, when R and NSE are ≥ 0.5, the model has acceptable performance (Moriasi et al., 2007). In addition, Germann and Zawadzki (2002) suggested that the threshold of predictability is 1/e≒0.37, assuming that the CSI values follow the exponential law. According to the standard, the results of Table 4 show that the machine learning-based models performed well as precipitation nowcasting models (R >0.8, NSE >0.5, CSI >0.5)."*

Reference
*Moriasi, D. N., Arnold, J. G., Van Liew, M. W., Bingner, R. L., Harmel, R. D., & Veith, T. L.: Model evaluation guidelines for systematic quantification of accuracy in watershed simulations. Trans. ASABE, 50(3), 885-900, 2007.*
*Germann, U. and Zawadzki, I.: Scale-Dependence of the Predictability of Precipitation from Continental Radar Images. Part I: Description of the Methodology, Mon. Weather Rev., 130, 2859–2873, https://doi.org/10.1175/1520-0493(2002)130<2859:SDOTPO>2.0.CO;2, 2002.*

- L. 283-285: "our model performs better than U-Net in predicting peak precipitation [...] prediction

accuracy of the ConvLSTM model for maximum precipitation was higher than that of our model" - please confirm these statements by adequate metrics, not from visual inspection

➔ We have added a few more verification metrics (FSS, PSD) to support our discussion:

L405: *"The results can also be confirmed through the FSS of each model (Fig. 6). When comparing Rad-cGAN and U-net, as lead time and rainfall intensity increase, both models decrease FSS; however, Rad-cGAN model exhibited superior performance. However, ConvLSTM had a relatively high FSS value under high rainfall intensity compared to those of the other two models..*

L423: *"Figure 8 shows the PSD for each result in Fig. 7. Based on Fig. 7, all models exhibited a blurring effect compared to the ground truth. However, when comparing U-net and Rad-cGAN, Rad-cGAN has slightly lower blurring effect. This is because a sharper image can be generated when cGAN is applied to the U-net structure (Isola et al. (2017)), which shows that the cGAN technique was successfully applied by our model."*

[Figure]

*Figure 6: Fraction Skill Scores (FSS) of model predictions at lead time of 10, 30, and 60 min at Soyang-gang Dam Basin. Panels from left to right express FSS of Rad-cGAN, U-net, ConvLSTM.*

[Figure]

*Figure 8: Radially averaged power spectral density (PSD) at forecasting time t = 23 August 2018, 17:50 UTC, for model predictions and observation*

- Ll. 287 ff: what is the basis for normative statements such as "good" and "sufficient"?

➔ As in the response of previous comment, we state that the model has good and sufficient performance when CSI, R, are greater than 0.5 by considering references. We have added the number of metrics to support our statement in the revised manuscript.

- Why use an entirely different presentation format in Fig. 5 to evaluate the performance in the catchments? Anyway, a revised version of the paper should not evaluate model performance for the dam locations and the catchments.

➔ As in the response of previous comment about verification on dam site and basin, we have added more explanation about our main objective.

L74: "*The aim of the present study was to develop an advanced precipitation nowcasting model for multiple dam basins that can be applied as an early warning system. The decision-making process at upstream dams with regard to flood control, which is directly related to urban and rural water management, influences flood risk considerably. From such a dam management perspective, water level and precipitation prediction at dam sites are major factors to be considered, suggesting that increasing rainfall prediction accuracy at the dam site is essential for effective flood management. To develop an advanced precipitation nowcasting model with good prediction performance for dam sites and basins in general, we designed a model based on the cGAN approach (Rad-cGAN) for multiple dam domains of the Soyang-gang, Andong, and Chungju dam basins in South Korea.*"

To clarify this, we have revised the title as follows:

"*Rad-cGAN v1.0: Radar-based precipitation nowcasting model with conditional Generative Adversarial Networks for multiple dam domains*"

**[RC2]**

We thank the reviewers for their constructive comments on our manuscript. In the following paragraphs, the reviewers' comments are in black font and our point-by-point responses are in blue.

This work adapts several classical CNN-based deep learning (DL) models for precip nowcasting. Specifically, the authors used Rad-cGAN, an adversarial learning network, to perform short-term prediction using radar data. The authors also demonstrated transfer learning, which is interesting. I think the work is publishable after some moderate revision.

1. L100, "For image translation tasks …" Here the objective of G should be minimizing the whole L_{cGAN}, not just the second part of it.

➔ We have corrected as follows:

L116: "*For image translation tasks, when $G$ is trained to produce a targeted image (y) from input (x) with random noise (z), the objective of $D$ will try to maximize the loss function $\mathcal{L}_{cGAN}(G,D)$ while $G$ will try to minimize $\mathcal{L}_{cGAN}(G,D)$.*"

2. Lookback period was fixed to 3 steps (i.e, t-30, t-20, and t-10). We know recurrent neural networks can be sensitive to length of lookbacks. Have you tried longer lookback?

➔ As per reviewer's suggestion, we have added results of model trained with longer lookback, with 5 steps (i.e., t-50, t-40, t-30, t-20, and t-10).

L182: "*First, we compared the results using a total of four and six consecutive radar reflectivity images to determine the input historical data length. As a result of 10 min precipitation prediction at the Soyang-gang Dam site, in the case of CSI (at the rainfall intensity of 0.1 mm h-1), the case of using six historical data was slightly better than case of using four data, but in R, RMSE, and NSE, the results of using four data were better. Through this, samples that consisted of four consecutive radar reflectivity images (t-30, t-20, t-10 min, and t) and the image at t+10 min were selected.*"

3. Model domain is fixed to 128x128. To me, this is really inconvenient. What if you need to deal with large model domains? Can you apply the 128x128 model to other nearby areas?

➔ We used 128 × 128 domain that can cover whole-dam basins, because our main purpose was to develop a rainfall prediction model to improve prediction skill for dam sites and watersheds. In addition, in terms of computational cost and computer sources, it was judged that it was inefficient to train the model using excessive area data.

Although a model was developed for the dam basin, the model can be used for other regions if the spatial dimension is 128 × 128 due to the model structure.

4. Resolution of all figures need to be improved. The fonts are all blurry. Suggest regenerate the figure using DPI>=300.

➔ We have improved figure quality.

5. Figure 3. I don't see a significant advantage of Rad-cGAN over Unet and Conv-LSTM. All the deep learning models underestimated the high magnitude rainfall events, which leads to my next comment.

➔ As we mentioned in the conclusion, the underestimation trends in the machine-learning-based model is a persisting issue; however, what we have shown is that applying the cGAN method can increase predictability compared to using only the U-net structure.

6. Here the authors only considered autoregression using radar reflectivity. Do you think incorporating other features (e.g., DEM, Wind, temp) can improve prediction skill?

➔ We also expect that the performance will be improved when topographical and meteorological factors that affect rainfall are added, and as we have written in the conclusion. We    expect to confirm this in future research.

**[RC3]**

We thank the reviewers for their constructive comments on our manuscript. In the following paragraphs, the reviewers' comments are in black font and our point-by-point responses are in blue.

**Summary:**

This paper is concerned with the prediction of precipitation at high temporal and spatial resolution and short lead-time commonly referred to as precipitation nowcasting. The authors explore the use of conditional generative adversarial networks (cGAN) to generate auto-regressive predictions of rain from radar images (1 km resolution) up to 90 min. The authors compare their results to several baselines from the deep learning field on data collected in sub-regions of Korea. They also explore several fine-tuning strategies to transfer the learning from one region to another.

Overall, the paper tackles an interesting problem where traditional methods such as NWP do not perform well. Given the growing availability of observation data, I expect that deep learning approaches will become more popular in this field. Therefore, this study, and especially its exploration of transfer learning, is timely. However, I have several major comments that require clarifications from the authors before the manuscript can be recommended for publication.

**Major comments:**

1. This paper should provide more context on the existing literature and on how it contributes to the field of nowcasting. In particular Ravuri et. al. 2021 also investigates the use of conditional GAN for precipitation nowcasting with extensive (probabilistic) evaluation. Could the authors further discuss how their method differs from Ravuri et. al.?

➔ As pointed out by reviewer, we have added following paragraph about differences between our work and the work of Ravuri et al. (2021) to clarify our intentions.

L74: *"The aim of the present study was to develop an advanced precipitation nowcasting model for multiple dam basins that can be applied as an early warning system. The decision-making process at upstream dams with regard to flood control, which is directly related to urban and rural water management, influences flood risk considerably. From such a dam management perspective, water level and precipitation prediction at dam sites are major factors to be considered, suggesting that increasing rainfall prediction accuracy at the dam site is essential for effective flood management.*

*To develop an advanced precipitation nowcasting model with good prediction performance for dam sites and basins in general, we designed a model based on the cGAN approach (Rad-cGAN) for multiple dam domains of the Soyang-gang, Andong, and Chungju dam basins in South Korea."*

L127: *"In this study, we developed a radar-based precipitation nowcasting model using a cGAN framework. Recently, research on weather prediction using cGAN, an advanced machine learning approach, has been conducted extensively (e.g., Rüttgers et al., 2019; Ravuri et al., 2021). For example, Ravuri et al. (2021) proposed a generator consisting of two modules; conditioning stack (using CNN to extract representation of input); and sampler (using ConvGRU to generate prediction). The model, which used ConvGRU, could observe spatiotemporal changes of inputs such as ConvLSTM, and attempted to improve performance by extracting features from different spatial dimensions and deriving the results. Whereas the generator used to predict future radar map, the discriminator used a dual architecture that distinguishes the real and generated frames, to ensure both temporal and spatial consistency. Unlike the model proposed by Ravuri et al. (2021), our model adopts a U-net architecture that uses a CNN layer in image generation based on the underlying Pix2Pix model; the architecture exhibits outstanding performance in image-to-image translation tasks (Isola et al., 2017).    Also, we considered only spatial consistency in the PatchGAN discriminator, which distinguishes images for each N × N patch (N can be smaller than the full size of the image). U-net-based precipitation nowcasting model has previously demonstrated performance superior to that of a traditional radar-based precipitation nowcasting model that uses optical flow (Ayzel et al., 2020). Therefore, here, we apply the basic cGAN methodology to the U-net structure to improve performance and confirm the applicability of the transfer learning methodology to multiple domains."*

2. One of the main claims of the paper is to outperform other baselines on a new dataset. To support that claim, the authors need to provide stronger evidences in the form of extensive evaluation:

1. Higher thresholds: The case made for nowcasting is to help with risk management. However, CSI is computed based on the 0.1mm/h threshold, which is pretty low to have any practical impact on risk management. Could the authors add evaluation at higher thresholds. For instance, I suggest adding the thresholds used in Ravuri et. al. 2021. While predicting intense events is more challenging, it is a more informative measure of progress for the field of nowcasting.

2. Additional metrics: Previous studies have also established a set of relevant metrics that should

be included in this study. For instance, PSD plots at various lead-times as a proxy for blurriness of the prediction, and skill scores over aggregated regions (e.g. 2x2, 4x4, 8x8 pixels). Note that these metrics are likely to pick up on positive characteristics of GAN predictions (e.g. sharp predictions and spatial consistency).

➔ As per reviewer's suggestion, we have added the CSI for higher rainfall intensity thresholds. Also, we had added fraction skill score (FSS) and power spectral density (PSD) to evaluate model result in the revised manuscript. When calculating CSI and FSS, we use intensity thresholds of 0.1, 1.0, 4.0, and 8.0 mm/h, and when calculating FSS, we use neighborhood sizes of 1, 5, 15, and 25 km.

L298: *"We used the CSI (Eq. (8)), which is a measure of categorical forecast performance, to verify the model accuracy for precipitation event detection.*

$$CSI = \frac{hits}{hits + false\ alarms + misses}$$

(8)

*where hits (correct event forecasts), false alarms (incorrect event forecasts), and misses (missed events) are defined by a contingency table (Table 2). Also, FSS can spatially verify model performance by comparing fraction of grid points of prediction and ground truth, which exceed certain rainfall intensity thresholds within the neighborhood (Eq. (9)).*

$$FSS = 1 - \frac{\sum_{i=1}^{n}(P_p - P_o)^2}{\sum_{i=1}^{n}P_p{}^2 + \sum_{i=1}^{n}(P_o)^2}$$

(9)

*where $P_p$ and $P_o$ are the fractions of prediction and observation, respectively, calculated by specific thresholds in neighborhood size. For calculating CSI and FSS, we selected several intensity thresholds, including 0.1, 1.0, and 5.0 mm $h^{-1}$, and for FSS, we used neighborhood sizes of 1, 5, and 15 km. Additionally, we calculated the radially averaged power spectral density (PSD) of predictions and observations to assess blurring effect of predicted image by models."*

L405: *"The results can also be confirmed through the FSS of each model (Fig. 6). When comparing Rad-cGAN and U-net, as lead time and rainfall intensity increase, both models decrease FSS; however, Rad-cGAN model exhibited superior performance. However, ConvLSTM had a relatively high FSS value under high rainfall intensity compared to those of the other two models."*

L423: "*Figure 8 shows the PSD for each result in Fig. 7. Based on Fig. 7, all models exhibited a blurring effect compared to the ground truth. However, when comparing U-net and Rad-cGAN, Rad-cGAN has slightly lower blurring effect. This is because a sharper image can be generated when cGAN is applied to the U-net structure (Isola et al. (2017)), which shows that our model successfully applied the cGAN technique*"

[Figure]

Figure 6: Fraction Skill Scores (FSS) of model predictions at lead time of 10, 30, and 60 min at Soyang-gang Dam Basin. Panels from left to right express FSS of Rad-cGAN, U-net, ConvLSTM.

[Figure]

*Figure 8: Radially averaged power spectral density (PSD) at forecasting time t = 23 August 2018, 17:50 UTC, for model predictions and observation*

3. Probabilistic evaluation: One of the main advantages of training a generative model, such as a GAN, is the ability to generate multiple predictions and conduct probabilistic evaluation on forecast ensembles. This leads to the following questions:

1. Does Rad-cGAN generate multiple samples?

➔ The output of model depends on input data size. Therefore, if input data are multiple samples (i.e., input data size = (No. of sample, 128, 128, 4)), the output are also multiple samples (i.e., output data size = (No. of sample, 128, 128, 1)). This has been added in the revised manuscript.

L265: "*The model was trained using data from the summers (June–August) of 2014–2017 and its precipitation nowcasting capacity assessed using data from the summer of 2018. To predict radar reflectivity data 10 min ahead, four latest radar reflectivity data (t-30, t-20, t-10 min, and t min; t being the forecast time) were used as input data. The model can generate multiple samples (No. of samples, 128, 128, 1) corresponding to the number of samples of the past four consecutive input data (No. of samples, 128, 128, 4).*"

2. If so, how are deterministic metrics such as CSI computed? Are the multiple samples aggregated into an average before computing the metrics? Is a single sample used for evaluation?

3. If not, why? Given that the motivation for this work is risk management which relies on assessing risks integrated over the full distribution of possible events, this would seem to be of critical importance.

➔ As the input samples are generated continuously over time, the order of the output samples is the same as the data period. To evaluate model for dam site, we calculated all metrics of one pixel (dam site location) and then averaged them over the data period (No. of samples). Also, to evaluate the model for dam basin, all metrics were calculated for each pixel of a dam basin and averaged down to the total number of pixels. Then, the calculated metrics of each sample are averaged over the data period (No. of samples). We have added this information about calculation of metrics in the revised manuscript.

L309: "*To calculate each verification metric, in the case of the evaluating for dam sites, we calculated all metrics for one pixel (dam site location) in the dam basins and averaged them over the data period (No. of samples). Also, in the case of evaluating dam domain, all metrics for each pixel in the dam basins were calculated and averaged over the data period (No. of samples).*"

4. Baselines: Please include PySTEPS (publicly available) as a baseline for your evaluation. Note that to make PySTEPS a competitive baseline, it might need to be fed more context than 128x128, as this is an advection method which is going to be more penalized at the boundaries of the prediction than DL methods.

➔ As per reviewer's suggestion, we are currently using PySTEPS to predict future precipitation based on our test dataset (summer of 2018), and the results will be added as a benchmark model to evaluate model performance.

L211: "*2.3.1 PySTEPS*
*PySTEPS (Pulkkinen et al., 2019) is the open-source and community-driven python framework for radar-based probabilistic precipitation nowcasting that is considered a strong baseline model (Imhoff et al., 2019; Ravuri et al., 2021). In the present study, STEPS (Short-Term Ensemble Prediction System) (Bowler et al., 2006) nowcast ensemble from the pySTEPS library was used as the benchmark model.*
*To generate precipitation predictions, we first provided input precipitation images (unit: dBR) that were transformed from four consecutive radar reflectivity images (from t-30 to t) based on a Z-R relationship (Eq. (1)). The transformed precipitation was also used to estimate motion field, and*

*the motion field and precipitation were used as input data in the STEPS model. Future precipitation at a lead time of up to 90 min for the test period (JJA of 2018) was generated based on the average across 20 ensemble members from the results of STEPS nowcasts. The source code of pySTPES is available in GitHub (https://pysteps.github.io, last accessed: 5 April 2022)."*

5. Data leakage and meta-optimization: The split between the different datasets needs to be clarified. It is now common practice to divide the available data between training set (for optimization), validation set (for hyper-parameter tuning and best checkpoint selection) and test set (for final evaluation). While the training set and test set are clearly defined, I did not find any reference to the validation set. This begs the questions:

   1. On which dataset were the various hyper-parameters tuning done?

➔ For hyperparameter tuning, we used 2014–2016 (June–August) radar data and 2017 (June–July) data to train the model of each combination. Then, 2017 (August) data were used to calculate mean absolute error (MAE) and CSI (at an intensity threshold of 0.1 mm/h) to obtain the optimal combination of hyperparameters. We have added this information in the revised manuscript.

L173: *"Before proceeding with training to optimize the model for the input data, hyperparameter tuning is required to determine the most optimal model structure and training settings. We selected the following hyperparameters: number of layers, number of hidden nodes, convolution filter size, patch size, batch size, and learning rate. To select the appropriate hyperparameter combination, the model for each combination was trained using radar data from 2014 to 2016 (June to August) and data from 2017 (June to July). Subsequently, using data from 2017 (August), the mean absolute error (MAE) and critical success index (CSI) (at an intensity threshold of 0.1 mm/h) were calculated to obtain the optimal combination of hyperparameters. Based on the tuning results, the MAE range was 0.45–47.66 and the CSI range was 0.0–0.83, and the results confirmed that hyperparameters influence model performance considerably. Based on the combinations that performed optimally, we determined the model structure and training settings."*

   2. Was the early stopping metric defined on the training set or the test set?

➔ The early stopping metric was defined as the generator loss based on 100 validation samples (not participating in training) randomly sampled from the training dataset. We have added this information in the revised manuscript.

L195: *"To achieve the optimal model, an early stopping technique that stops the training model when the loss stops improving was applied. The loss metric was defined as the generator loss based on 100 validation samples randomly sampled from the training dataset that was not used to train the model."*

3. How have the baselines been tuned?

➔ U-net has hyperparameters (No. of layers, No. of hidden nodes, convolution filter size) similar to those of Rad-cGAN, and are tuned based on the following hyperparameters: batch size and learning rate. Also, in the case of ConvLSTM, the number of layers, nodes, batch size, learning rate is tuned. The data splitting procedures for model tuning and evaluation was are the same as those for Rad-cGAN. We have added this information in the revised manuscript.

L235: *"The hyperparameters of the ConvLSTM model (i.e., number of layers, number of nodes, convolution filter size, batch size, and learning rate) were tuned using a procedure similar to that applied in Rad-cGAN (Sect. 2.2.4)."*

L257: *"As the reference model, hyperparameters for the U-net structure (number of layers, number of nodes, and convolution filter size) were set to be equivalent to those of Rad-cGAN (Sect. 2.2.2), and hyperparameters related to training settings (batch size and learning rate) were tuned using procedures similar to those of Rad-cGAN (Sect. 2.2.4). To optimize the model, $L1$ loss and ADAM optimizers were used as in the case of ConvLSTM (Sect. 2.3.2). The model was trained using 600 epochs with early stopping and the batch size set to 8."*

3. The second claim of the paper is to successfully employ transfer learning techniques to generalize to different regions. In the machine learning context, transfer learning is often used to transfer knowledge acquired on a large dataset to a similar but smaller dataset. While reducing the computational cost of training the model is definitely of interest, it is the limited availability of data

for the new task (here the new region) that usually justifies doing transfer learning rather than training from scratch. In the context of this paper, it is not clear why transfer learning is useful: there is the same amount of data for the new regions, and most of the data is disregarded at train time (not included in the crop). Several alternative strategies should be included as baselines:

1. Pre-trained model on first region, without fine-tuning (effectively case 2, since optimizing the discriminator without fine-tuning the generator is like using the pretrained prediction model without modification)

2. Pre-trained model on first region, with fine-tuning (generator+discriminator) on the new region (close to case 3)

3. Pre-trained model on random crops from all the data available (the 11 radars), without fine-tuning

4. Pre-trained model on random crops from all the data available (the 11 radars), with fine-tuning on the new region

5. Training model on new region from scratch (case 1)

Additionally, as a methodological demonstration, the authors should also consider running experiment 3b) with varying amount of data from the new region (for instance, using the equivalent of 1, 2 or 3 summers). This would be an informative experiment (and more realistic use case) on the amount of data required to do transfer learning.

➜ As per reviewer's suggestion, we have revised our transfer learning plan, as shown in the table below.

L333: *"We used transfer learning to train our model for different dam basins, i.e., Andong and Chungju, with a pre-trained model that was completely trained by data from the Soyang-gang Dam Basin. In addition, in existing papers that have successfully applied the transfer learning strategies, it was used to develop a model for a new domain using a pre-trained model based on vast data. Consequently, we used the pre-trained model with Daecheong Dam, Juam Dam, and Yongdam Dam basin data, in addition to Soyang-gang Dam data, to assess the amount of data required to develop a model for a new dam domain. The selected strategies were inspired by a previous approach of transferring GAN (Wang et al., 2018; Mo et al., 2020). We formulated two strategies for each pre-trained model. First, the weights of the pre-trained generator were frozen and used directly in new dam domain (Case 2, 4). Next, to the weights of the pre-trained generator were fine-tuned (1/10$^{th}$ of*

*the original learning rate) and the discriminator trained (Case 3, 5). In addition, the entire model was trained for the new domain (Case 1) (Table 3a). The model was trained for the Chungju and Andong Dam domains, separately, using the five strategies (Table 3b)."*

*Table 3: Experimental design for transfer learning strategies to train model with different domain. (a) Detailed training procedure of each strategy, (b) data used to train the model according to each strategy.*

| (a) Training strategies | | |
|---|---|---|
| No. | Generator | Discriminator |
| Case 1 | Train from the scratch | Train from the scratch |
| Case 2 | Use pre-trained parameters for one domain | - |
| Case 3 | Fine tuning pre-trained parameters for one domain* | Train from the scratch |
| Case 4 | Use pre-trained parameters for multiple domains | - |
| Case 5 | Fine tuning pre-trained parameters for multiple domains* | Train from the scratch |

| (b) Training dataset | | | |
|---|---|---|---|
| | Pre-trained domain | Andong Dam domain | Chungju Dam domain |
| Case 1 | - | 2014–2017 (JJA) at Andong Dam domain | 2014–2017 (JJA) at Chungju Dam domain |
| Case 2 | 2014-2017 (JJA) at Soyang-gang Dam domain | - | - |
| Case 3 | 2014-2017 (JJA) at Soyang-gang Dam domain | 2014–2017 (JJA) at Andong Dam domain | 2014–2017 (JJA) at Chungju Dam domain |
| Case 4 | 2014-2017 (JJA) at Soyang-gang/Daecheong/Juam/Yongdam Dam domain | - | - |
| Case 5 | 2014-2017 (JJA) at Soyang-gang/Daecheong/Juam/Yongdam Dam domain | 2014–2017 (JJA) at Andong Dam domain | 2014–2017 (JJA) at Chungju Dam domain |
| * Use 1/10th of original learning rate | | | |

**Minor comments:**

- Why not work with the precipitation itself, rather than reflectivity? The non-linearity may

translate in different error amplifications for different precipitation amounts.

➔ We trained models using raw radar data to generate radar reflectance maps and converted them into rainfall using empirical equations (Z-R relationships). This is because well-converted radar-based rainfall data could not be obtained from the data source (KMA). In addition, we decided that it would be better to predict radar reflectivity itself, considering the uncertainty of rainfall converted into an empirical formula, and the possibility of using an advanced conversion formula in the future.

- Please provide more information about the datasets, this includes (per data split):
  - Distribution of precipitation amount
  - Number of examples

➔ As per reviewer's comment, we have added detailed information about datasets in Table 1 as follows:

*Table 1: Distribution of precipitation amount and number of examples of (a) train dataset, and (b) test dataset of each dam domains and sites.*

| (a) Train dataset | | | | | | |
|---|---|---|---|---|---|---|
| Interval in mm h-1 | Soyang-gang Dam Basin | Soyang-gang Dam site | Andong Dam Basin | Andong Dam site | Chungju Dam Basin | Chungju Dam site |
| $0 \leq R < 0.1$ | 84.63 | 85.00 | 83.04 | 83.57 | 84.81 | 85.03 |
| $0.1 \leq R < 1.0$ | 10.23 | 10.19 | 11.56 | 11.02 | 10.75 | 10.61 |
| $1.0 \leq R < 4.0$ | 3.78 | 3.35 | 3.91 | 3.89 | 3.43 | 3.30 |
| $4.0 \leq R < 8.0$ | 0.92 | 1.03 | 0.94 | 0.98 | 0.68 | 0.69 |
| $8.0 \leq R < 10.0$ | 0.18 | 0.21 | 0.20 | 0.23 | 0.13 | 0.16 |
| $10.0 \leq R$ | 0.27 | 0.23 | 0.35 | 0.32 | 0.20 | 0.21 |
| No. of examples | 27,905 examples | | 29,136 examples | | 29,691 examples | |
| (b) Test dataset | | | | | | |
| Interval in mm h-1 | Soyang-gang Dam Basin | Soyang-gang Dam site | Andong Dam Basin | Andong Dam site | Chungju Dam Basin | Chungju Dam site |
| $0 \leq R < 0.1$ | 90.77 | 91.92 | 87.41 | 87.73 | 86.54 | 87.61 |

| | | | | | | |
|---|---|---|---|---|---|---|
| 0.1 ≤ R < 1.0 | 5.77 | 5.16 | 7.59 | 7.48 | 8.63 | 8.06 |
| 1.0 ≤ R < 4.0 | 2.65 | 2.26 | 3.77 | 3.92 | 3.80 | 3.58 |
| 4.0 ≤ R < 8.0 | 0.58 | 0.42 | 0.87 | 0.54 | 0.76 | 0.51 |
| 8.0 ≤ R < 10.0 | 0.10 | 0.06 | 0.15 | 0.09 | 0.12 | 0.13 |
| 10.0 ≤ R | 0.14 | 0.17 | 0.21 | 0.24 | 0.14 | 0.11 |
| No. of examples | 9,753 examples | | 6,598 examples | | 6,137 examples | |

This is particularly important to verify if there is enough data to make any assessment about certain events (e.g. high intensity precipitation)

- Why just consider the summer? Is there more precipitation during those months?

➔ The reason lies in the characteristics of Korea's research area that we selected. Since Korea has a seasonality in precipitation, most of the precipitation is concentrated in summer. Therefore, to reduce the tendency of underestimation due to data imbalance in machine learning, we used only summer rainfall, where high intensity rainfall occurs frequently. This has been added in the revised manuscript.

L105: *"We selected the available radar reflectivity data in summer (June–August, JJA) from 2014 to 2018 considering high intensity rainfall occurs in summer due to rainfall seasonality, a characteristic of our study domain."*

- How was the test set normalized? Using min max of training set? Or the test set? This is important to make sure there is no data leakage, and the predictions are not using information from the future.

➔ We used min max of training dataset to normalize both the training and test datasets. We have added the relevant information in the manuscript, as follows:

L107: *"For rapid and effective training, the raw radar reflectivity data (dBZ) were converted to grayscale (0–255), and the data range was scaled to 0–1 using the Min-Max scaler method (min-max values from training dataset)."*

- The transfer learning part is missing a case, fine-tuning both generator and discriminator.

➔ As we wrote in the conclusion, in addition to the methods used, there are various transfer learning techniques that can be applied to the GAN methodology. The methods are expected to be added through future research.

- Please provide all the plots and figures at higher resolution.

➔ We have revised and improved the quality of images in the revised manuscript.

- For transfer learning case 2, what is the use of training a discriminator if the generator is pre-trained and frozen? The predictions will be unaffected by the fine-tuning. It seems to me that Case 2 is equivalent to just applying the model trained on the first region to a new one.

➔ There was a mistake in the original manuscript about the explanation of case 2 strategies. Case 2 had weights similar to those of the pre-trained model. This has been corrected in the revised manuscript.

- For evaluation, please include tables at 90 min lead-time rather than 10 min. Could you also include mean quantities. If you keep reporting the median (which is more computationally intensive as you need to keep track of the whole distribution), please include other percentiles (95%, 99%) to give a sense of the uncertainty.

➔ We selected 10-min predictions at dam site because we thought it is necessary to show quantitative indicators of the results after 10 min when evaluating the model's performance, since the model was trained to generate 10 min prediction images. Therefore, we have showed performance at higher lead times in Fig. 4, separately.

As per the reviewer's suggestion, we have revised the results and used boxplots for evaluation.

[Figure]

*Figure 5: Box plot of verification metrics of model predictions at the lead time up to 90 min over all grid cells from the Soyang-gang Dam region. Left panels from top to bottom represent R, RMSE, NSE, and right panels from top to bottom represent CSI at intensity threshold of 0.1, 1.0, 5.0 mm h-1.*

- Line 69. Please clarify what is proposed on top of the cited paper. If using the technique as is, please state "we apply the transfer learning [...]".

➔ We have corrected it.

 L83: "*We applied the transfer learning technique (Pan and Yang, 2009) [...]*"

- Line 78. Typo.

➔ We have corrected it.

 L91: "[...] *Andong Dam Basin (D3) areas [...]*"

- Paragraph 2.2.1. This paragraph is hard to read. Could you please rewrite it in a more precise way.

➔ We have rewritten the paragraph in the revised manuscript.

- I would suggest using "discriminator" and "generator" for the two submodels of the GAN rather than "discriminative model", and "generative model" which have a broader meaning.

➔ We have corrected it.

- Line 95. Do you mean "It consists of a generator (G) that produces the distribution [...]"?

➔ We have corrected it.

L112: "*It consists of a generative model (G) that produces the distribution of real data from random noise, […]*"

- Line 122. Do you mean "To prevent overfitting"?

➔ We have corrected it.

L148: "*To prevent overfitting, a dropout layer with a rate of 0.5 […]*"

- Paragraph 2.2.3. This paragraph is hard to read, in part because the word input is used to refer to two different quantities. Could you please rewrite it in a more precise way.

➔ We have rewritten the paragraph in the revised manuscript.

- Line 136. What does the following mean? "The size of the patch (N) was determined by the structure of the entire discriminative model, and it increased as the model became deeper. We constructed a discriminator model through optimization with a 34 × 34 patch size." How was N decided? Using hyper-parameter optimization? By "constructing a discriminator through optimization", do you mean training the weights of the model or optimizing the architecture?

➔ Since the patch size is defined by structure of discriminator, we tuned the following hyperparameters: No. of layers, size of convolution filter, strides, paddings.

To avoid any confusion, we have added following sentence.

L159: *"This patch represents the receptive field, which is the region in the input image that is used to measure the associated feature of the output layer. Consequently, the size of the patch (N) was determined based on the structure of the entire discriminator (e.g., number of layers, nodes, filter size, paddings, and strides), and it increased as the model became deeper. We constructed a discriminator model with a 34 × 34 patch size through hyperparameter tuning."*

- Line 257. This statement is a bit misleading. Fine-tuning is not defined by changing the learning rate to 1/10 of the original setup. It typically uses a smaller learning rate to make small adjustments to the weights. The computational savings come from the lower number of training steps for the fine-tuning rather than the lower learning rate.

➔ We have corrected it as follows:

L325: "*Fine-tuning uses a smaller learning rate (e.g., ~1/10th of the original learning rate) and is one of the most effective ways to transfer knowledge.*"

- Line 314. I am not sure how this shows that baselines are properly tuned and trained.

➔ We stated that our results had CSI similar or greater than those of previous works. This is to show that we did not put less effort on the basic model compared to the Rad-cGAN model we developed.

- Line 342. How come case 2 (generator trained on region 1 and not fine-tuned on new region) performs better than case 1 which trains (from scratch) on the new region with the same amount of data as the pretraining on region 1? This is a very surprising result. Or am I misunderstanding what case 2 is doing?

➔ As pointed out in the previous comment, as the reviewer said, it is right to directly use the previous model for the new domain in case 2. In our previous manuscript, we mistakenly discussed that the

additional training process of the discriminator affects the performance of case 2, which is superior to case 1. As a result of a new discussion on this, we suggest that the cause of the result may be the influence of hyperparameter tuning.

In case 1, we did not tune the hyperparameters for the new domain. The pre-training model, in which hyperparameter tuning was completed for the Soyang-gang Dam Basin (D1), was confirmed to perform well if other domains had similar types of data, because the model was optimized and generalized through the validation dataset. However, even if the new data is similar to the pre-trained data, the generalization and optimization of the model (Case1) were less comprehensive compared when compared with that of the pre-trained model, resulting in relatively poor performance.

We have added more discussion on this comment after applying the revised transfer learning strategy to the revised manuscript.

L442: *"Hyperparameter tuning would have had a significant impact on the results where Case 2 performs better than Case 1. Unlike the pre-training model, which confirmed that model optimization and generalization were completed through the hyperparameter tuning process, in case 1, we did not proceed with hyperparameter tuning for the new domain. Although the new domain has properties similar to those of the previous domain, minor changes in hyperparameters also result in differences in performance, so that optimization and generalization of the model (Case 1) were less comprehensive than in the pre-training model, resulting in relatively poor performance."*

- Please include a reference to a quantitative metric (plot or table) for any statement about performance.

➔ We have corrected it.

- Fig 6. Please also include samples of predictions (not just the error) for different lead-time for all the models/baselines.

➔ We have corrected Fig. 6, as follows:

[Figure]

*Figure 7: Precipitation observation example at forecasting time t = 23 August 2018, 17:50 UTC, for model predictions and (a) ground truth (OBS). Panels from top to bottom express ground truth: (b) prediction of Rad-cGAN model, (c) prediction of U-net based model, (d) prediction of ConvLSTM, and € prediction of pySTEPS*

---

## Author Response (AR2)

We thank the reviewers for their valuable comments on the manuscript. In the following paragraphs, the reviewers' comments are in black font, and our point-by-point responses are in blue.

**Major comments**

**Presentation quality**

The quality of the figures is unacceptable, mostly due a lack of resolution. Some figures are just impossible to interpret (e.g. Fig. 8). I pointed out this issue in my first report, and I have to say I am quite annoyed that it has not been addressed. Instead, it has become worse. The recommendation to use colormaps that take into account colour-blindness was ignored, too.

Furthermore, the manuscript needs considerable language editing, and the line of arguments is often difficult to follow. I recommend putting much more emphasis into the readability and conciseness of the main text.

➔ As suggested by the reviewer, we have improved the quality of the all figures. We have also modified the colors of the figures to be color-blind friendly based on www.ColorBrewer.org

[Figure]

*Figure 1: (a) Composite map of radar reflectivity and location of the dam basins; (b) selected areas over the dam basin. D1, D2, and D3 represent the areas of Soyang-gang, Chungju, and Andong Dam Basins, respectively. Maps were created using ArcGIS software by Esri; Base-map source: Esri, HERE, Garmin, © OpenStreetMap contributors, and the GIS User Community.*

[Figure]

*Figure 3: Box plot of the verification metrics of model predictions at the lead time up to 90 min over all grid cells from the Soyang-gang Dam region. From top to bottom, left panels represent R, RMSE, and NSE, and right panels represent the CSI at intensity thresholds of 0.1, 1.0, and 5.0 mm h⁻¹. Grey dotted line represents the predictability threshold (1/e≒0.37).*

[Figure]

*Figure 5: Example of precipitation at forecasting time t = 23 August 2018, 17:50 UTC, for model predictions and (a) ground truth (OBS). Panels from top to bottom express ground truth: (b) prediction of Rad-cGAN model, (c) prediction of U-net based model, (d) prediction of ConvLSTM, and (e) prediction of pySTEPS.*

[Figure]

*Figure 6: Radially averaged power spectral density (PSD) at forecasting time t = 23 August 2018, 17:50 UTC, for model predictions and observation.*

[Figure]

*Figure 8: Box plot of the verification metrics of model predictions at lead time up to 90 min over all grid cells from the Andong Dam Basin. (a) R, (b) NSE, (c) RMSE, and (d, e, f) CSI at intensity thresholds of 0.1, 1.0, and 5.0 mm h⁻¹, respectively. Grey dotted line represents the predictability threshold (1/e ≒0.37).*

[Figure]

*Figure 9: Box plot of the verification metrics of model predictions at the lead time up to 90 min over all grid cells from the Chungju Dam Basin. (a) R, (b) NSE, (c) RMSE, and (d, e, f) CSI at intensity thresholds of 0.1, 1.0, and 5.0 mm h$^{-1}$, respectively. Grey dotted line represents the predictability threshold (1/e ≒0.37).*

**Spatial verification set-up**

I am not convinced by the author's arguments; from a dam management perspective, it is almost irrelevant how much precipitation falls at the location of the dam. The relevant parameter is the water level in the reservoir, which depends on the inflow, which in turn depends on the rainfall over the reservoir catchment. To be more explicit, I suggest dropping the entire part of the study that is related to this issue.

➜ As per the reviewer's suggestion, we have revised our manuscript to focus on the dam basin. We modified the values in Tables 4 and 5, which demonstrate the verification metrics for the dam sites to average values for the entire dam basin, and modified the line plots to box plots representing the values of the entire area in pixels.

*Table 4: Comparison of the average values of the verification metrics for a 10-min precipitation prediction of different models at the Soyang-gang Dam Basin during summer (June–August), 2018.*

|  | R | RMSE (mm h$^{-1}$) | NSE | CSI (0.1 mm h$^{-1}$) | CSI (1.0 mm h$^{-1}$) | CSI (5.0 mm h$^{-1}$) |
|---|---|---|---|---|---|---|

| | | | | | | |
|---|---|---|---|---|---|---|
| **Rad-cGAN** | **0.7891** | **0.6138** | **0.5367** | **0.7827** | **0.6262** | 0.1772 |
| U-Net | 0.7822 | 0.6626 | 0.4582 | 0.7783 | 0.5688 | 0.0793 |
| ConvLSTM | 0.6976 | 0.6508 | 0.4694 | 0.7247 | 0.5462 | 0.2019 |
| pySTEPS (baseline) | 0.7076 | 0.7826 | 0.4100 | 0.7181 | 0.5803 | **0.3214** |
| Persistence (baseline) | 0.5839 | 0.8117 | 0.1678 | 0.6821 | 0.4987 | 0.2197 |

*Table 5: Comparison of the average values of the verification metrics for a 10-min precipitation prediction of the five different models using different transfer learning strategies for the (a) Andong and (b) Chungju Dam Basins in the summer (June–August) of 2018.*

| **(a) Andong Dam domain** | | | | | | |
|---|---|---|---|---|---|---|
| | R | RMSE (mm h$^{-1}$) | NSE | CSI (0.1 mm h$^{-1}$) | CSI (1.0 mm h$^{-1}$) | CSI (5.0 mm h$^{-1}$) |
| Case 1 | 0.7945 | 0.8169 | 0.4926 | 0.7662 | 0.6073 | 0.1193 |
| Case 2 | 0.8037 | 0.7673 | 0.5624 | 0.7756 | 0.6482 | 0.1523 |
| Case 3 | 0.8146 | 0.7858 | 0.5351 | 0.7916 | 0.6067 | 0.1317 |
| Case 4 | 0.7952 | 0.7407 | 0.5948 | 0.7782 | 0.6497 | 0.2399 |
| Case 5 | 0.7952 | 0.7319 | 0.6051 | 0.7794 | 0.6472 | 0.2682 |
| **(b) Chungju Dam domain** | | | | | | |
| | R | RMSE (mm h$^{-1}$) | NSE | CSI (0.1 mm h$^{-1}$) | CSI (1.0 mm h$^{-1}$) | CSI (5.0 mm h$^{-1}$) |
| Case 1 | 0.7909 | 0.9221 | 0.5161 | 0.7893 | 0.6169 | 0.1639 |
| Case 2 | 0.7863 | 0.8609 | 0.5876 | 0.7831 | 0.6492 | 0.2981 |
| Case 3 | 0.7995 | 0.8849 | 0.5623 | 0.7920 | 0.6351 | 0.2324 |
| Case 4 | 0.7776 | 0.8808 | 0.5661 | 0.7761 | 0.6380 | 0.2614 |
| Case 5 | 0.7809 | 0.8783 | 0.5685 | 0.7803 | 0.6386 | 0.2657 |

[Figure]

*Figure 8. Box plot of the verification metrics of model predictions at the lead time up to 90 min over all grid cells from the Andong Dam Basin. (a) R, (b) NSE, (c) RMSE, and (d, e, f) CSI at intensity thresholds of 0.1, 1.0, and 5.0 mm h⁻¹, respectively. Grey dotted line represents the predictability threshold (1/e ≒0.37).*

[Figure]

*Figure 9. Box plot of the verification metrics of model predictions at the lead time up to 90 min over all grid cells from the Chungju Dam Basin. (a) R, (b) NSE, (c) RMSE, and (d, e, f) CSI at intensity thresholds of 0.1, 1.0, and 5.0 mm h⁻¹, respectively. Grey dotted line represents the predictability thresholds (1/e ≒0.37).*

**Lack of a competitive benchmark**

I welcome the use of PySTEPS as a benchmark model. However, I think that the used verification metrics cannot be simply applied to a probabilistic ensemble forecast. Can you just compute the ensemble mean, and then treat it the same way you would treat a deterministic forecast? I don't think so. Please refer e.g. to Imhof et al. (2020) how to treat probabilistic PySTEPS forecasts in a verification context, or replace the ensemble forecast by a deterministic PySTEPS model. Otherwise, I suspect that the performance of the benchmark model might not be assessed fairly.

Looking at Fig 7, I am also very concerned about the role of edge effects in the verification of PySTEPS. Why not use a larger spatial window for the prediction to avoid this?

Overall, I am quite surprised by the poor performance of PySTEPS in comparison to other studies, specifically with regard to the CSI and FSS. What could be the reason?

➔ We have revised the benchmark model to S-PROG (Spectral Prognosis) from the pySTEPS library, which is a deterministic nowcasting model.

L218: *"2.3.1 PySTEPS*

*PySTEPS (Pulkkinen et al., 2019) is an open-source and community-driven Python framework for radar-based deterministic and probabilistic precipitation nowcasting, and is considered a strong baseline model (Imhoff et al., 2019; Ravuri et al., 2021). In this study, deterministic S-PROG (Seed, 2003) nowcast from the pySTEPS library was used as the benchmark model.*

*To predict precipitation, we input the precipitation images (unit: dBR) transformed from four consecutive radar reflectivity images (from t-30 to t), which were the same as the input of the Rad-cGAN model, based on the Z-R relationship (Eq. (1)). Additionally, the transformed precipitation was used to estimate the motion field, which was used together with precipitation as input data in the model. Future precipitation at a lead time of up to 90 min for the test period (JJA, 2018) was generated from the results of the S-PROG nowcasts. The source code of pySTEPS is available at GitHub repository (https://pysteps.github.io, last accessed: 23 May 2022)."*

We discussed the poor prediction performance of pySTEPS to be the reason for not predicting the rainfall area at the edge of the domain, as observed from the results of the typhoon event. As per the reviewer's suggestion, to reduce this edge effect, we have added the results of pySTEPS using 384×384 pixels input data, which was extended by 128 pixels on each side of the original input data. However, owing to limited data availability in the real-world, it was inappropriate to compare its performance with that of Rad-cGAN using extended data only for pySTEPS; hence, the edge-effecteliminated pySTEPS results were evaluated against Rad-cGAN that was trained using the same extended data (384×384 pixels).

L111: *"Furthermore, to reduce the edge effect caused by the fast Fourier transform (FFT), which is used for scale decomposition of pySTEPS (S-PROG) nowcast (Pulkkinen et al., 2019; Foresti and Seed, 2014), we derived the pySTEPS results using 384 × 384 km2 input data extended by 128 pixels on each side of the original input data (128 × 128 pixels)."*

L412: *"PySTEPS shows poor performance (Fig. 5) compared to previous studies (e.g., Imhoff et al., 2020) in the verification metrics (Table. 4 and Figs. 3-4). The overall prediction performance degrads particularly because the precipitation area near the edge of the basin is not predicted. To better understand this side effect, we reran pySTEPS and Rad-cGAN with the extended data of 384×384 pixels. Compared to the predictions in Fig. 5, the typhoon event (Fig. 7) shows that using the extended area reduces the edge effect of pySTEPS and properly maintains high rainfall intensity, thereby improving the performance. Moreover, the average R and CSI (at the highest rainfall intensity of 5.0 mm h-1) for the 10-min precipitation prediction during the entire test period is calculated as 0.77 and 0.38, respectively, indicating that the performance improves quantitatively compared to the previous results (R=0.70 and CSI=0.32). Additionally, the prediction performance of typhoon event improves using the extended area in Rad-cGAN (Fig. 7), and the average R and CSI (at the rainfall intensity of 5.0 mm h-1) in the 10-minute rainfall forecast for the entire test period improves from 0.79 to 0.80 and from 0.18 to 0.37, respectively. Both models show improved performance using extended area, but considering the applicability of the model to real-world problems with limited data availability, we conclude that, unlike pySTEPS, Rad-cGAN is more efficient in rainfall prediction without considering the edge effects of the spatial size of the input domains."*

[Figure]

*Figure 7. Example of precipitation at forecasting time t = 23 August 2018, 17:50 UTC, for model prediction using increased input area (384 × 384). Panels from top to bottom express (a) prediction of Rad-cGAN model, and (e) prediction of pySTEPS.*

**Model software and reproducibility**

Improvements were made, but the repository does not yet contain the PySTEPS model implementation.

➜ We added the PySTEPS implementation to our repository.

Code and data availability: "*Source code of the model architecture is available at the GitHub repository https://github.com/SuyeonC/Rad-cGAN (last access: 13 June 2022). The pre-trained model for Soyang-gang Dam Basin and example data are available at https://doi.org/10.5281/zenodo.6460012. The radar reflectivity composite data samples provided by the Korea Meteorological Administration (KMA) are available at the public service: https://data.kma.go.kr/resources/html/en/ncdci.html (last access: 11 November 2021). The dataset for the entire period can be obtained through a separate request to the KMA.*"